# A BREGMAN PROXIMAL VIEWPOINT ON NEURAL OPERATORS

## ABSTRACT

We present several advances on neural operators by viewing the action of operator layers as the minimizers of Bregman regularized optimization problems over Banach function spaces. The proposed framework allows interpreting the activation operators as Bregman proximity operators from dual to primal space. This novel viewpoint is general enough to recover classical neural operators as well as a new variant, coined Bregman neural operators, which includes the inverse activation function and features the same expressivity of standard neural operators. Numerical experiments support the added benefits of the Bregman variant of Fourier neural operators for training deeper and more accurate models.

## 1 INTRODUCTION

Neural operators (Kovachki et al., 2021; 2023), a recent extension of neural networks, have emerged as a versatile framework for learning mappings between function spaces. These operators have shown great potential in solving partial differential equations (PDEs) and simulating complex dynamical systems. The exploration of neural architectures for the approximation and learning of operators has led to the development of a variety of models.

One influential contribution is the Fourier Neural Operator (FNO) (Li et al., 2021a), sketched in Figure 1, which transforms encoded input data into frequency components in order to learn intricate relationships in the frequency domain. More recently, the Group-Equivariant FNO (G-FNO) (Helwig NEW et al., 2023) additionally leverages symmetries to design equivariant Fourier layers, thereby enhancing the representation power and robustness of the architecture. To better scale the depth of neural operators, the F-FNO (Tran et al., 2023) proposed separable spectral layers and improved residual connections, along with a bag of training tricks. The FNO are extended to Wavelet Neural Operators (WNO) (Tripura & Chakraborty, 2023) by replacing Fourier layers with wavelet layers to further exploit multiscale information. The U-shaped Neural Operator (U-NO) (Rahman et al., 2023) adapts the U-net architecture for neural operators, enabling mapping between function spaces through integral operators, thus broadening the applicability of neural architectures to diverse domains. Differently, the DeepONet architecture (Lu et al., 2021) comprises two intertwined components: a branch network responsible for encoding discrete input function spaces, and a trunk network dedicated to encoding the domain of output functions. Operating as a conditional model, DeepONet leverages the embedding of inputs and outputs via a dot product operation, facilitating the approximation of complex functions through a structured network topology. Finally, Neural Inverse Operators (NIO) (Molinaro et al., 2023) tackle inverse problems by combining DeepONet and FNO architectures to map operators to functions, thereby extending the applicability of neural operators to coefficient estimation tasks.

Some approaches inspired by attention mechanisms, pivotal in image and natural language processing, have also been considered in operator learning. LOCA (Learning Operators with Coupled Attention) (Kissas et al., 2022) facilitates robust gradient estimation, particularly in scenarios with limited training data, by combining attention with kernel mechanisms. The General Neural Operator Transformer (GNOT) (Hao et al., 2023) is a scalable framework based self-attention mechanisms allowing to deal with heterogeneous inputs useful for modeling diverse physical systems.

Some physics-informed variants integrating information from PDEs during the learning process have been proposed enhancing model interpretability and generalization: PI-DeepONet (Wang et al., 2021) and its Long-Time Integration variant (LTI-PI-DeepONet) (Wang & Perdikaris, 2023), PINO (Physics-Informed Neural Operator) (Li et al., 2021b) a hybrid extension of FNO, or other variations such as V-DeepONet (Goswami et al., 2022) and Modified DeepONet (Wang et al., 2022).

Figure 1: Illustration of the $t$-th layer of Fourier Neural Operators. The upper branch applies a linear transformation $R_t$ to the Fourier modes using the Fourier transform $\mathcal{F}$ and its inverse $\mathcal{F}^{-1}$. The lower branch performs an affine transformation in the latent space.

NEW

**Contributions.** Unlike previous works (Kovachki et al., 2021), which directly consider the compositional form of neural operators, our approach introduces a distinct perspective by formulating the action of each operator layer as the minimizer of a regularized optimization problem over functions. This optimization connects the current hidden representation to the next, with the choice of a regularization implicitly defining the activation operator through the lens of the Bregman proximity operator. Our expressive framework not only convers existing neural operators but also introduces a novel variant, termed *Bregman neural operator*, which demonstrates improved predictive performance as its depth increases. Its applicability is grounded by universal approximation results proven for sigmoidal-type activation operators. Beyond its unifying aspect and its ability to design novel neural operators, the proposed framework allows applying the extensive body of literature on proximal numerical optimization, of which Bregman proximity operators belong to, in order to study neural operators. This opens the way to extend the analysis done on neural networks to (Bregman) neural operators in the same spirit of Combettes & Pesquet (2020a;b).

FIX

**Outline.** The rest of the paper is organized as follows: Section 2 is dedicated to the presentation of definitions and background knowledge on neural operators and Bregman proximity operator. In Section 3, we introduce the operator layers as the solution of a functional optimization problem. In addition, we show that this new mapping allows recovering the classical neural operators and creating a more general family of so-called Bregman neural operators. In Section 4, we provide a preliminary universal approximation result for Bregman neural operators. Finally, in Section 5, we conduct an extended experimental study comparing on benchmark datasets our Bregman variant with the classical FNO.

## 2  BACKGROUND AND DEFINITIONS

Here, we introduce some definitions required for the understanding of the rest of the paper as well as the necessary background on neural operators and Bregman proximity operator. We will use basic concepts from convex analysis such as subdifferential, $\Gamma_0$ space and Fenchel conjugate, whose definitions are recalled in Appendix A.

NEW

### 2.1  OPERATOR LEARNING: APPLICATION TO LEARNING THE SOLUTION MAP OF PDES

Operator learning finds significant applications in the context of PDEs in order to efficiently approximate solutions to PDEs without the need to solve them repeatedly from scratch (Li et al., 2021b; Serrano et al., 2023; Raonic et al., 2023). Given a nonempty bounded open set $D \subset \mathbb{R}^d$, and some time horizon $\tau > 0$, we consider the generic family of PDEs over $D \times [0, \tau]$ of the form

$$F_a\big((\partial^\alpha u(x,t))_{\alpha \in \mathbb{N}^{d+1}, |\alpha| \le k}\big) = f(x,t) \text{ on } D \times ]0, \tau] \text{ and } \begin{cases} u(x,0) = u_0(x) \text{ on } D, \\ u(x,t) = u_b(x,t) \text{ on } \partial D \times ]0, \tau], \end{cases}$$

(1)

where $F_a$ is a possibly nonlinear partial differential operator, $f$ denotes a source term, $u_b$ is a boundary condition, $u_0$ is an initial condition, and $u : D \to \mathbb{R}^n$ is the solution of the PDE.

The problem we will tackle in our numerical section is the *initial value problem*. This involves finding the oracle mapping $\mathcal{G}$ from any initial condition function $u_0$ to the solution $u(\cdot, \bar{\tau})$ of the PDE at a certain time horizon $\bar{\tau} \in ]0, \tau]$.

More generally, the oracle operator $\mathcal{G}$ could be a mapping between two different function spaces $\mathcal{A}$ and $\mathcal{U}$. Without loss of generality, given some bounded open sets $D \subset \mathbb{R}^d$, with $d \in \mathbb{N}_+$, we let $\mathcal{A} = \mathcal{A}(D, \mathbb{R}^n)$ and $\mathcal{U} = \mathcal{U}(D, \mathbb{R}^k)$, with $n, k \in \mathbb{N}_+$, be some separable Banach spaces of functions. For instance, $\mathcal{A}$ can represent the spaces of continuous functions from $D \to \mathbb{R}^n$. Hereafter, $\mathcal{A}$ and $\mathcal{U}$

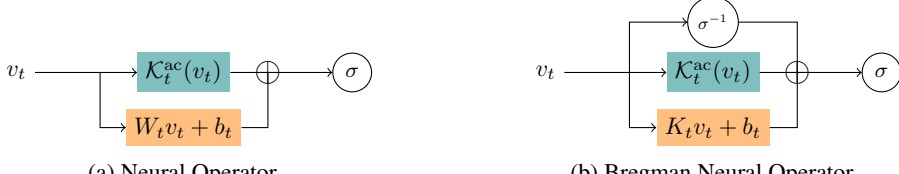

(a) Neural Operator.  (b) Bregman Neural Operator.

Figure 2: Illustration of the $t$-th layer of (Bregman) Neural Operators. On the left, the identity term and the linear term $K_t v_t + b_t$ have been merged into $(I + K_t)v_t = W_t v_t$.

will be referred to as the spaces of *input functions* and *output functions*, respectively. In a nutshell, operator learning consists in finding the unknown ground-truth correspondence operator $\mathcal{G} \colon \mathcal{A} \to \mathcal{U}$ given $N \in \mathbb{N}_+$ pairs of input-output functions $\{a_i, u_i\}_{i=1}^N$.

## 2.2 Neural Operators

Among the existing models to parametrize an approximation of $\mathcal{G}$, we focus on neural operators, which are parametric mappings $\mathcal{N} \colon \mathcal{A} \to \mathcal{U}$ of the form

$$(\forall a \in \mathcal{A}), \quad \mathcal{N}(a) = \mathcal{Q} \circ \mathcal{L}_T \circ \ldots \circ \mathcal{L}_1 \circ \mathcal{P}(a), \tag{2}$$

where

- $\mathcal{P} \colon \mathcal{A}(D, \mathbb{R}^n) \to \mathcal{A}(D, \mathbb{R}^{n_0})$ is a local *lifting operator* mapping the input function to its first hidden representation;
- $\mathcal{Q} \colon \mathcal{U}(D, \mathbb{R}^{n_T}) \to \mathcal{U}(D, \mathbb{R}^k)$ is a local *projection operator* mapping the last hidden representation to the output function;
- For every $t \in \{1, \ldots, T\}$, $\mathcal{L}_t \colon \mathcal{V}_{t-1}(D_t, \mathbb{R}^{n_{t-1}}) \to \mathcal{V}_t(D_t, \mathbb{R}^{n_t})$ is an *operator layer* where each $D_t \subset \mathbb{R}^{d_t}$ is an open bounded set, $\mathcal{V}_t = \mathcal{V}_t(D_t, \mathbb{R}^{n_t})$ is a suitable Banach space of functions such that $\mathcal{V}_0 = \mathcal{A}(D, \mathbb{R}^{n_0})$ and $\mathcal{V}_T = \mathcal{U}(D, \mathbb{R}^{n_T})$, for consistency.
- Each component of the neural operator (2) depends on a finite dimensional parameter. Collectively those parameters constitute a vector $\theta \in \Theta \subset \mathbb{R}^p$.

Most methodological developments in neural operators have focused on tailoring the operator layers $\mathcal{L}_1, \ldots, \mathcal{L}_T$ to specific application. Traditionally, their design mirrors standard neural networks, replacing finite-dimensional linear layers with integral linear operators in function spaces and interpreting activation functions as Nemytskii operators that apply nonlinear transformations pointwise. When the input spaces $D_t$ are the same throughout the layers and equals $D$, a popular class of operator layers, sketched in Figure 2a, is of the form    FIX

$$\mathcal{L}_t(v_t) = \sigma(W_t v_t + \mathcal{K}_t^{\mathrm{ac}}(v_t) + b_t), \tag{3}$$

where $W_t \in \mathbb{R}^{n_t \times n_{t-1}}$ is a matrix, $b_t \in \mathbb{R}^{n_t}$ is a bias vector and $\sigma$ is a *local* nonlinear map acting pointwise from $\mathbb{R}^{n_t}$ to $\mathbb{R}^{n_t}$. Moreover, we have a *non-local* linear operator $\mathcal{K}_t^{\mathrm{ac}} \colon L^2(D, \mathbb{R}^{n_{t-1}}) \to L^2(D, \mathbb{R}^{n_t})$. In its simplest version, $\mathcal{K}_t^{\mathrm{ac}}$ is an integral kernel operator of the form $(\mathcal{K}_t^{\mathrm{ac}}(v))(x) = \int_D k_t(x, y)v(y)dy$, for all $x \in D$, with $k_t$ being a kernel to be specified (Kovachki et al., 2023).    FIX
Specific examples include those based upon a convolution performed in the Fourier space (Li et al., 2021a; Kovachki et al., 2021), a graph kernel network (Anandkumar et al., 2020) or its multipole variant (Li et al., 2020) to name a few.
Hereafter, we follow a different path and propose to interpret operator layers from the viewpoint of a proximal optimization by seeing the parametric form of (3) as the minimizer of a Bregman regularized optimization problem. This novel perspective allows us to propose a *novel architecture*, displayed in Figure 2b, of the form

$$\mathcal{L}_t(v_t) = \sigma(\sigma^{-1}(v_t) + K_t v_t + \mathcal{K}_t^{\mathrm{ac}}(v_t) + b_t), \tag{4}$$

involving an additional nonlinear term $\sigma^{-1}(v_t)$, and where $K_t \in \mathbb{R}^{n_t \times n_{t-1}}$ is a matrix. In this formulation, when all the weights are zero, then $\mathcal{L}_t$ is the identity operator. A similar architecture    FIX
was originally proposed in Frecon et al. (2022) in the finite dimensional setting. Extending this work to neural architectures acting on Banach function spaces requires addressing non-trivial mathematical

challenges. These include defining operator layers rigorously, particularly the proper formulation of Legendre functions on function spaces, the associated Bregman divergence, and the Bregman proximity operator. In the next section, we formalize these notions, laying the groundwork for the proposed novel perspective on neural operators. The reader interested in the technical details is invited to refer to Appendix A.

NEW

### 2.3 BREGMAN PROXIMITY OPERATOR

The definition of Bregman proximity operator relies on the choice of a Bregman divergence, loosely called distance, which itself is built upon a Legendre function (see, e.g., Rockafellar (1970)).

**Definition 1** (Legendre function). *A function $\phi\colon \mathbb{R}^n \to ]-\infty, +\infty]$ is called* Legendre *if it is proper convex lower semicontinuous and satisfies the following properties:* i) $\mathrm{int}(\mathrm{dom}\,\phi) = \mathrm{dom}\,\partial\phi$ *and* $\partial\phi$ *is single-valued on its domain;* ii) $\phi$ *is strictly convex on* $\mathrm{int}(\mathrm{dom}\,\phi)$.

In the finite dimensional setting, Legendre functions $\phi$ are typically built from an elementary Legendre function $\varphi\colon \mathbb{R} \to ]-\infty, +\infty]$ as $\phi\colon x \in \mathbb{R}^n \to \sum_{i=1}^{n} \varphi(x_i)$. Since here we stand in an infinite dimensional setting, i.e., Lebesgue function space, the counterpart of the previous finite sum structure is a convex integral functional defined below. Also, we will allow vector valued functions.

**Fact 1** (Convex integral functionals on Lebesgue spaces based on Legendre function). *Let $D \subset \mathbb{R}^d$ be an open bounded set. Let $p, q \in [1, +\infty]$ be conjugate exponents, that is such that $1/p + 1/q = 1$, and set $\mathcal{V} := L^p(D, \mathbb{R}^n)$ and $\mathcal{V}^* = L^q(D, \mathbb{R}^n)$. The spaces $\mathcal{V}$ and $\mathcal{V}^*$ can put in duality via the pairing $\mathcal{V} \times \mathcal{V}^* \to \mathbb{R}, \quad (v, u) \mapsto \langle v, u \rangle = \int_D \langle v(x), u(x) \rangle dx$. Let $\phi \in \Gamma_0(\mathbb{R}^n)$ be a Legendre function and let $\Phi\colon \mathcal{V} \to ]-\infty, +\infty]$ be such that*

$$\Phi(v) = \int_D \phi(v(x))dx. \tag{5}$$

*Then $\Phi \in \Gamma_0(\mathcal{V})$, $\mathrm{dom}\,\partial\Phi = \{v \in \mathcal{V} \,|\, \text{for a.e. } x \in D, v(x) \in \mathrm{int}(\mathrm{dom}\,\phi) \text{ and } (\nabla\phi) \circ v \in \mathcal{V}^*\}$, $\partial\Phi$ is single valued on $\mathrm{dom}\,\partial\Phi$, and, for every $v \in \mathrm{dom}\,\partial\Phi$, $\partial\Phi(v) = \{\nabla\phi \circ v\}$. The unique element $\nabla\phi \circ v$ of $\partial\Phi(v)$ will be denoted by $\tilde{\nabla}\Phi(v)$, suggesting it will serve as a kind of gradient of $\Phi$ at $v$*[1]

The integral functional $\Phi$ in (5) inherits certain properties of $\phi$, such as $p$-uniform convexity — an extension of strong convexity when $p = 2$. This characteristic, proved in Proposition 4 of the appendix, will play a pivotal role in Remark 1. Additionally, we have the property detailed in Remark 7 of the appendix.

We are now equipped to define Bregman distances in Lebesgue spaces. We recall that the concept of Bregman divergence was introduced by Bregman in the pioneering work of (Bregman, 1967) in the context of alternating projection methods. It provides a generalization of a kind of distance measure, such as the Euclidean distance, which finds application in statistics and machine learning to quantify notably the difference between distributions.

NEW

**Definition 2** (Bregman distance in Lebesgue spaces). *Under the notations of Fact 1, the* Bregman distance *with respect to $\Phi$ is defined as*

$$D_\Phi\colon \mathcal{V} \times \mathcal{V} \to [0, +\infty], \quad D_\Phi(u, v) = \begin{cases} \Phi(u) - \Phi(v) - \langle u - v, \tilde{\nabla}\Phi(v) \rangle & \text{if } v \in \mathrm{dom}\,\partial\Phi \\ +\infty & \text{otherwise}. \end{cases}$$

Finally, we can define the Bregman proximity operator (Nguyen, 2017), which extends the (Euclidean) proximity operator, widely used in optimization. The Euclidean proximity operator itself generalizes projections by replacing the indicator function of a convex set with appropriate convex functions. For additional details, the reader can refer to Bauschke & Combettes (2017).

**Definition 3** (Bregman proximity operator). *Let $\mathcal{V} = L^p(D, \mathbb{R}^n)$ with $p \in [1, +\infty[$. Let $g \in \Gamma_0(\mathcal{V})$ and let $\Phi \in \Gamma_0(\mathcal{V})$ be defined as in Fact 1, with $\phi \in \Gamma_0(\mathbb{R}^n)$ be Legendre and such that $\mathrm{ran}\,\partial(\Phi + g) = \mathcal{V}^*$. Then the* Bregman proximity operator *of $g$ relative to $\Phi$ is defined as*

$$\mathrm{prox}_g^\Phi\colon \mathcal{V}^* \to \mathcal{V}, \quad v^* \mapsto \mathrm{argmin}\left\{\langle \cdot, -v^* \rangle + \Phi + g\right\}.$$

FIX

---

[1]Note that in general the domain of the function $\Phi$ has empty interior, so Gâteaux and/or Frechet differential cannot be properly defined.

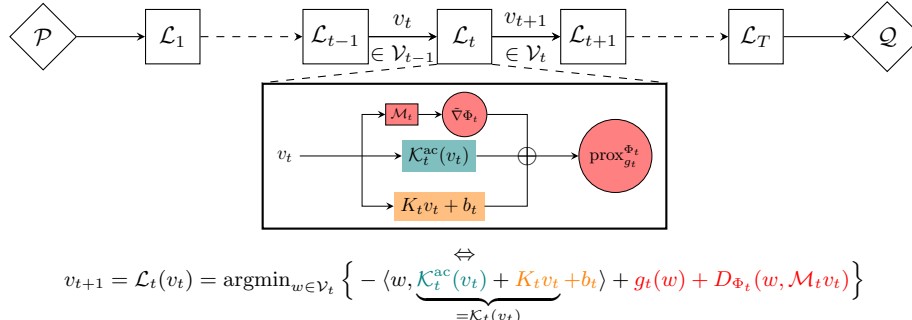

$$v_{t+1} = \mathcal{L}_t(v_t) = \operatorname*{argmin}_{w \in \mathcal{V}_t} \Big\{ - \langle w, \underbrace{\mathcal{K}_t^{\mathrm{ac}}(v_t) + K_t v_t + b_t}_{= \mathcal{K}_t(v_t)} \rangle + g_t(w) + D_{\Phi_t}(w, \mathcal{M}_t v_t) \Big\}$$

Figure 3: Illustration of the Bregman proximal viewpoint on operator layers. The action of each operator layer is viewed as the minimizer of the regularized optimization problem where each term in the objective can be linked to a part of the architecture, as evidenced by the color code.

NEW

Note that $\operatorname{prox}_g^\Phi$ is well-defined since $\Phi + g$ is strictly convex and lower semicontinuous and $\operatorname{ran} \partial(\Phi + g) = \mathcal{V}^*$, and it holds $\operatorname{prox}_g^\Phi = [\partial(\Phi + g)]^{-1}$.

**Remark 1.**

(i) *If $\mathcal{V} = L^p(D, \mathbb{R}^n)$ with $p \in {]1, +\infty[}$, the condition $\operatorname{ran} \partial(\Phi + g) = \mathcal{V}^*$ is satisfied if $\phi$ is $p$-uniformly convex (see Proposition 4 in the appendix). Moreover, by Remark 7, if $p = 1$ and $\operatorname{dom} \phi^* = \mathbb{R}^n$, then $\operatorname{ran} \partial \Phi = \mathcal{V}^*$.*

(ii) *If instead of $\operatorname{ran} \partial(\Phi + g) = \mathcal{V}^*$, one asks the stronger condition $\operatorname{ran}(\partial \Phi + \partial g) = \mathcal{V}^*$, then we have $\partial(\Phi + g) = \partial \Phi + \partial g$ and the Bregman proximity operator writes down as $\operatorname{prox}_g^\Phi = (\partial \Phi + \partial g)^{-1}$ and $\operatorname{ran}(\operatorname{prox}_g^\Phi) \subset \operatorname{dom} \partial \Phi$.*

## 3 REVISITING NEURAL OPERATORS

In Section 3.1, we propose a novel Bregman proximal viewpoint on operator layers. Then, we establish several connections. First, we show in Section 3.2 that the proposed framework is general enough to recover most classical operator layers when the Legendre function $\phi$ is the Euclidean distance. Second, we show in Section 3.3 how it yields a new variant of neural operators when $\phi$ defines a general Bregman divergence. Finally, we apply our framework to Fourier neural operators in Section 3.4.

### 3.1 BREGMAN PROXIMAL VIEWPOINT ON OPERATOR LAYERS

Departing from usual kernel-based points of view (Kovachki et al., 2021), we suggest defining operator layers as the solution of functional optimization problems. For every $t = 1, \cdots, T$, $\mathcal{L}_t \colon \mathcal{V}_{t-1} \to \mathcal{V}_t$,

$$\mathcal{L}_t(v) = \operatorname*{argmin}_{w \in \mathcal{V}_t} \Big\{ -\langle w, \mathcal{K}_t(v) + b_t \rangle + g_t(w) + D_{\Phi_t}(w, \mathcal{M}_t v) \Big\} = \operatorname{prox}_{g_t}^{\Phi_t} \big( \tilde{\nabla} \Phi_t(\mathcal{M}_t v) + \mathcal{K}_t(v) + b_t \big),$$

(6)

where

- $\Phi_t \colon \mathcal{V}_t \to {]-\infty, +\infty]}$ is a convex integral functional on an appropriate Lebesgue space based on some Legendre function $\phi_t \in \Gamma_0(\mathbb{R}^{n_t})$, as defined in Fact 1. $D_{\Phi_t} \colon \mathcal{V}_t \times \mathcal{V}_t \to [0, +\infty]$ is the corresponding Bregman distance as detailed in Definition 2
- $\mathcal{M}_t \colon \mathcal{V}_{t-1} \to \mathcal{V}_t$ is a bounded linear operator which maps $\operatorname{dom} \partial \Phi_{t-1}$ into $\operatorname{dom} \partial \Phi_t$,
- $b_t \in \mathcal{V}_t^*$ and $\mathcal{K}_t \colon \mathcal{V}_{t-1} \to \mathcal{V}_t^*$ is a bounded linear operator of the form

$$\mathcal{K}_t(v)(x) = \int_{D_{t-1}} \kappa_t(x, dy) v(y),$$

with $\kappa_t \colon D_t \times \mathfrak{B}(D_{t-1}) \to \mathbb{R}^{n_t \times n_{t-1}}$ a *(transition) kernel* from $D_{t-1}$ to $D_t$, meaning a function which is measurable with respect to the first variable and a finite measure with respect to the second variable.

FIX

- $g_t \in \Gamma_0(\mathcal{V}_t)$ and $\mathrm{ran}(\partial\Phi_t + \partial g_t) = \mathcal{V}_t^*$.

Equation (6) is highly general, featuring an outer operation (the $\mathrm{prox}_{g_t}^{\Phi_t}$) and an inner operation (the $\tilde{\nabla}\Phi_t$), and can formally represent various layer architectures sketched in Figure 3. A key step in establishing this connection involves relating the proximity operator to activation operators. There are multiple ways to achieve this by varying the choice of the pair $(\Phi_t, g_t)$. In the following sections, we explore two specific choices for this pair, demonstrating how (6) recovers classical neural operators (3) (where $\tilde{\nabla}\Phi_t$ is the identity) and introduces a novel architecture (4), in which $\tilde{\nabla}\Phi_t$ acts as the inverse activation operator.

**Remark 2** (Form of linear operator $\mathcal{K}_t$). *Often in applications, the kernel of the linear operator $\mathcal{K}_t$ is split into two terms: an absolutely continuous part and a single pure point part, i.e., $\kappa_t = \kappa_t^{\mathrm{ac}} + \kappa_t^p$, where, for every $x \in D_t$, and measurable set $A \subset D_{t-1}$,* FIX FIX

$$\kappa_t^{\mathrm{ac}}(x, A) = \int_A k_t(x, y)dy \quad and \quad \kappa_t^p(A) = K_t \delta_{\varphi_t(x)}(A),$$

*with $k_t \colon D_t \times D_{t-1} \to \mathbb{R}^{n_t \times n_{t-1}}$, $K_t \in \mathbb{R}^{n_t \times n_{t-1}}$, $\varphi_t \colon D_t \to D_{t-1}$ measurable, and $\delta_{\varphi_t(x)}$ the delta Dirac at $\varphi_t(x) \in D_{t-1}$. Thus, we have* FIX

$$\mathcal{K}_t(v)(x) = \mathcal{K}_t^{\mathrm{ac}}(v)(x) + \mathcal{K}_t^{\mathrm{p}}(v)(x) = \int_{D_{t-1}} k_t(x, y)v(y)dy + K_t v(\varphi_t(x)).$$

**Remark 3** (Special case of identical domains). *The linear operator $\mathcal{M}_t$ should be chosen so that it maps $\mathrm{dom}\,\partial\Phi_{t-1}$ to $\mathrm{dom}\,\partial\Phi_t$. However, in (6), if the function $\phi_t$ does not depend on $t$ and all the domains $D_t$ are the same, then it is also true that the convex integral functional $\Phi_t$ does not depend on $t$ too. Then, we have $\mathrm{dom}\,\partial\Phi_{t-1} = \mathrm{dom}\,\partial\Phi_t$ and for the linear operator $\mathcal{M}_t$ we are allowed to choose the identity operator.*

**Remark 4.**

(i) *In view of Remark 1(ii), the condition $\mathrm{ran}(\partial\Phi_t + \partial g_t) = \mathcal{V}_t^*$ implies that $\mathrm{prox}_{g_t}^{\Phi_t} = (\partial\Phi_t + \partial g_t)^{-1}$ and hence $\mathrm{ran}(\mathrm{prox}_{g_t}^{\Phi_t}) \subset \mathrm{dom}\,\partial\Phi_t$. In this way $\mathrm{dom}\,\mathcal{L}_t = \mathcal{M}_t^{-1}(\mathrm{dom}\,\partial\Phi_{t-1})$ and $\mathrm{ran}(\mathcal{L}_t) \subset \mathrm{dom}\,\partial\Phi_t$ and the composition (2) is well-defined provided that for the lifting operator $\mathcal{P}$ it holds $\mathrm{ran}(\mathcal{P}) \subset \mathrm{dom}\,\partial\Phi_1$ (e.g., if $\mathcal{P}(v)(x) = \nabla\phi_1^*(Pv(x))$).*

(ii) *When $\mathcal{V}_{t-1} = \mathcal{V}_t$ and $\mathcal{M}_t$ is the identity, the operator layer (6) takes the form*

$$\mathrm{prox}_{g_t}^{\Phi_t}(\tilde{\nabla}\Phi_t(v) - \mathcal{B}_t v) = (\partial\Phi_t + \partial g_t)^{-1}(\tilde{\nabla}\Phi_t - \mathcal{B}_t)(v),$$

*where $\mathcal{B}_t \colon \mathcal{V}_t \to \mathcal{V}_t^*$. This is a* Bregman forward-backward operator, *which is well-known in the context of operator splitting methods in optimization (Nguyen, 2017; Bùi & Combettes, 2021).*

Concluding this section, we stress that as long as the couple $(\Phi_t, g_t)$ admits an explicit (closed form) Bregman proximity operator, this would define additional new types of operator layers. In Nguyen (2017), the author shows a number of examples (at the end of Section 2, from Example 2.9 to Example 2.12) of such couples that yield an explicit Bregman proximity operator. As a matter of fact, one may consider layers of type FIX

$$v \mapsto \sigma_2(\sigma_1^{-1}(v) + \mathcal{K}_t(v) + b_t),$$

with $\sigma_1$ being *strictly* monotone and $\sigma_2$ monotone, serving as activation operators appropriately coupled. Classical and Bregman neural operators emerge as special cases, where i) $\sigma_1 = \mathrm{Id}$ and $\sigma_2$ is any monotone function, for the former, and ii) $\sigma_1 = \sigma_2$ is *strictly* monotone, for the latter. Note that having $\sigma_1 = \sigma_2$ implies that the numerical implementation does not require to have an explicit form of $\sigma_1^{-1}$, as latter discussed in Remark 5 (ii). FIX

NEW

### 3.2 CLASSICAL NEURAL OPERATORS

Our first result, stated in the proposition below, unifies a broad class of classical neural operator layers through the prism of the optimization viewpoint of (6) when $D_{\Phi_t}$ is the Euclidean distance.

**Proposition 1** (Unifying classical neural operators). *Let $\mathcal{V}_t = L^2(D_t, \mathbb{R}^{n_t})$ be some Hilbert function space and $\Psi_t(v) = \int_{D_t} \sum_{i=1}^{n_t} \psi(v_i(x))dx$, where $\psi \in \Gamma_0(\mathbb{R})$ is a strongly convex Legendre function.* FIX

Table 1: Relationship between Legendre function $\psi$ and activation function $\psi^{*\prime}$.

| dom$\psi$ | $\psi$ | $\psi'$ | $\psi^*$ | $\psi^{*\prime}$ |
|---|---|---|---|---|
| $[-1, 1]$ | $t \mapsto -\sqrt{1-t^2}$ | $t \mapsto t/\sqrt{1-t^2}$ | $t \mapsto \sqrt{1+t^2}$ | ISRU |
| $[0, 1]$ | $t \mapsto t\log t + (1-t)\log(1-t)$ | $t \mapsto \log(\frac{t}{1-t})$ | $t \mapsto \log(1+e^t)$ | Sigmoid |
| $[-1, 1]$ | $t \mapsto \log(1-t^2) + t\operatorname{arctanh}(t)$ | $\operatorname{arctanh}$ | $\log\cosh$ | tanh |
| $[-1, 1]$ | $t \mapsto \sqrt{1-t^2} + t\arcsin(t)$ | $\arcsin$ | $-\cos$ | sin |
| $\mathbb{R}_{>0}$ | $t \mapsto \frac{1}{\beta^2}\operatorname{Li}_2(e^{-\beta t}) + \frac{1}{2}t^2$ | $t \mapsto \frac{1}{\beta}\log(e^{\beta t}-1)$ | $t \mapsto -\frac{1}{\beta^2}\operatorname{Li}_2(-e^{\beta t})$ | SoftPlus$_\beta$ |

*Consider the Euclidean distance defined from the elementary Legendre function $\phi_t = (1/2)|\cdot|^2 \in \Gamma_0(\mathbb{R}^{n_t})$ (see Section 2.3) and set $g_t = \Psi_t - (1/2)\|\cdot\|^2$. Then $g_t \in \Gamma_0(\mathcal{V}_t)$ and $\mathcal{L}_t$ defined in (6) acts between $L^2$ spaces as follows*

$$\mathcal{L}_t(v) = \operatorname{prox}_{\Psi_t - \frac{1}{2}\|\cdot\|^2}^{\frac{1}{2}\|\cdot\|^2}\big(\mathcal{M}_t v + \mathcal{K}_t(v) + b_t\big) = \nabla\Psi_t^*(\mathcal{M}_t v + \mathcal{K}_t(v) + b_t), \qquad (7)$$

*where $\nabla\Psi_t^* = (\psi^*)'(\cdot)$ matches a variety of monotone activation operators $\sigma$. In addition, when the domains are all the same, say $D_t = D$, $\mathcal{M}_t = I$, and the linear operator $\mathcal{K}_t = \mathcal{K}_t^{\mathrm{ac}} + \mathcal{K}_t^{\mathrm{p}}$ is as given in Remark 2, then $\mathcal{L}_t(v) = \nabla\Psi_t^*((I + K_t)v + \mathcal{K}_t^{\mathrm{ac}}(v) + b_t)$, where $(I + K_t)$ can be written as $W_t$. A schematic representation is reported in Figure 2a.*

In essence, Proposition 1 shows that the parametric structure of operator layers can be interpreted via the Bregman proximal operator, when the Bregman distance reduces to the Euclidean distance. The crucial aspect in establishing this connection is the observation that the Euclidean proximity operator of $g_t = \Psi - (1/2)\|\cdot\|^2$ simplifies to $\nabla\Psi^* = (\psi^*)'(\cdot)$, aligning with a broad spectrum of activation operators given an appropriate selection of $\psi$. We report in Table 1 the corresponding $\psi$ to retrieve several well-known activation operators. A proof concerning the characterization of the SoftPlus function is included in the appendix. To the best of our knowledge, $\nabla\Psi_t^*$ can only match monotonic activation operators, which notably discards GeLu and swish. To be more precise, Proposition 1 is general enough to deal with the broad class of activation functions that can be viewed as a proximity operators, which essentially boils down to any increasing 1-Lipschitzian function (see Proposition 2.3 in Combettes & Pesquet (2020a)). While this connection has been previously noted in the neural network literature (Combettes & Pesquet, 2020a; Frecon et al., 2022), our work extends this analysis to function spaces.

### 3.3 BREGMAN NEURAL OPERATORS

We now provide the counterpart of Proposition 1 for general Bregman distance.

**Proposition 2** (Designing Bregman neural operators)**.** *Let $\mathcal{V}_t = L^p(D_t, \mathbb{R}^{n_t})$ be some Lebesgue function space and $\Psi_t(v) = \int_{D_t} \sum_{i=1}^{n_t} \psi(v_i(x))dx$, where $\psi \in \Gamma_0(\mathbb{R})$ is a p-uniformly convex Legendre function ($\neq |\cdot|^2/2$). Consider the Bregman distance in function space defined from the elementary Legendre function $\phi_t(w) = \sum_{i=1}^{n_t} \psi(w_i)$ (see Section 2.3) and set $g_t = 0$. Then $\mathcal{L}_p$ defined in (6) acts between $L^p$ spaces as follows*

$$\mathcal{L}_t(v) = \operatorname{prox}_0^{\Psi_t}\big(\tilde{\nabla}\Psi_t(\mathcal{M}_t v) + \mathcal{K}_t(v) + b_t\big) = \nabla\Psi_t^*(\tilde{\nabla}\Psi_t(\mathcal{M}_t v) + \mathcal{K}_t(v) + b_t), \qquad (8)$$

*where $\nabla\Psi_t^* = (\psi^*)'(\cdot)$ matches a variety of monotone activation operators $\sigma$. In addition, when the domains are all the same, say $D_t = D$ and the linear operator $\mathcal{K}_t$ is of the form given in Remark 2, then we can take $\mathcal{M}_t = I$ and*

$$\mathcal{L}_t(v) = \nabla\Psi_t^*(\tilde{\nabla}\Psi_t(v) + K_t v + \mathcal{K}_t^{\mathrm{ac}}(v) + b_t). \qquad (9)$$

Concerning the operators $\nabla\Psi_t^* = (\psi^*)'(\cdot)$ and $\nabla\Psi_t^* = \psi'(\cdot)$, we stress that any of the $\psi$ listed in Table 1 are appropriate choices. Since $(\psi^*)'(\cdot)$ and $\psi'(\cdot)$ are inverse of each other, the layer of (9) boils down to

$$\mathcal{L}_t(v) = \sigma(\sigma^{-1}(v_t) + K_t v + \mathcal{K}_t^{\mathrm{ac}}(v) + b_t), \qquad (10)$$

where any invertible and monotone activation operator is allowed. Its schematic representation is reported in Figure 2b. This novel variant, called *Bregman Neural Operator* simply differs from

classical neural operators by the additional term involving the inverse activation operator. Finally, we note that the form of (9) corresponds to a mirror descent step (Nemirovskij & Yudin, 1983; Beck & Teboulle, 2003) with mirror map $\tilde{\nabla}\Psi_t$.

**Remark 5.**

(i) *When $K_t$, $\mathcal{K}_t^{\mathrm{ac}}$ and $b_t$ are zeros and $\mathcal{M}_t$ is the identity, then $\mathcal{L}_t$ reduces to the identity.*

(ii) *Concerning (10), we should ensure to feed the first layer with functions in $\mathrm{dom}\,\mathcal{L}_1$ as discussed in Remark 4 (i). This condition is for instance satisfied if $(\mathcal{P}v)(v) = \nabla\psi_1^*(Pv(x)) = \sigma(Pv(x))$. Note that in such situation, the inverse activation function does not need to have an explicit form. Indeed, when composing the different layers in (10), the inner inverse activation function will be cancelled out by the outer one.*

### 3.4 CASE OF FOURIER NEURAL OPERATORS

We study the implications of the proposed viewpoint in the peculiar case of Hilbert function spaces with equal input and output spaces, i.e., $\mathcal{V}_t = \mathcal{V}_t^* = L^2(D, \mathbb{R}^n)$ for every $t \in \{1, \dots, T\}$.

A popularly encountered scenario in practice is that where $D = \mathbb{T}^d$ is the unit torus and the kernel associated to the absolutely continuous part of $\mathcal{K}_t$ is translation invariant, i.e., $k_t(x, y) = k_t(x - y)$, thus indicating a convolution structure. Fourier operator layers (Li et al., 2021a) are then devised by leveraging the convolution theorem, stating that the action of $\mathcal{K}_t^{\mathrm{ac}}$ can be written as a linear operator in the Fourier domain:

$$\mathcal{K}_t^{\mathrm{ac}}(v)(x) = \int_D k_t(x - y)v(y)dy = \mathcal{F}^{-1}(R_t \cdot \mathcal{F}(v))(x), \qquad (11)$$

with $\mathcal{F}\colon L^2(\mathbb{T}^d, \mathbb{R}^n) \to \ell^2(\mathbb{Z}^d, \mathbb{R}^n)$ being the Fourier transform, $\mathcal{F}^{-1}$ its inverse, and $R_t \in \ell^2(\mathbb{Z}^2, \mathbb{R}^{n \times n})$. Often, $R_t$ does not range in the entire $\ell^2(\mathbb{Z}^2, \mathbb{R}^{n \times n})$ space but is parametrized by a finite parameter (Kovachki et al., 2023). It follows that the Bregman variant of Fourier operator layer reads $\mathcal{L}_t(v) = \sigma(\sigma^{-1}(v) + W_t v + \mathcal{F}^{-1}(R_t \cdot \mathcal{F}(v)) + b_t)$. The classical Fourier neural operator layer is retrieved by omitting the $\sigma^{-1}(v)$ term.

In this section, we addressed FNOs because they are widely used and simplify the analysis. In this respect, we note that we just specified the action of $\mathcal{K}_t^{\mathrm{ac}}$ by expressing it via direct and inverse Fourier series. So, in the end, it is only about finding efficient parametrizations, in some $\ell^p$ space, of linear integral operators between Lebesgue spaces. This has been achieved by using the Fourier transform, but in principle other transformations could be considered, provided we have an unconditional basis of the Lebesgue space of functions and an efficient way to compute the coefficients. For instance, the wavelet transform can be incorporated in Proposition 1 and Proposition 2 to retrieve WNOs (Tripura & Chakraborty, 2023) and their novel Bregman variant, respectively. In a nutshell, our framework is transparent to the parametrization of $\mathcal{K}_t^{\mathrm{ac}}$. FIX

## 4 EXPRESSIVITY OF BREGMAN NEURAL OPERATORS

In this section, we give a preliminary positive result concerning the universal approximation properties of Bregman neural operators.

In the following, the activation function $\sigma\colon \mathbb{R} \to I$ is required to be a homeomorphism between $\mathbb{R}$ and an open interval $I$ of $\mathbb{R}$ and of sigmoidal type, meaning that $\lim_{t \to -\infty} \sigma(t) = 0$ and $\lim_{t \to +\infty} \sigma(t) = 1$. Moreover, we assume that $\mathcal{A}$ and $\mathcal{U}$ are as follows

$$\mathcal{A}(D, \mathbb{R}^n) = \begin{cases} \mathcal{C}(\overline{D}, \mathbb{R}^n) \\ L^p(D, \mathbb{R}^n) \\ W^{m,p}(D, \mathbb{R}^n) \end{cases} \quad and \quad \mathcal{U}(D, \mathbb{R}^k) = \begin{cases} \mathcal{C}(\overline{D}, \mathbb{R}^k) \\ L^p(D, \mathbb{R}^k), \end{cases}$$

with $p \in [1, +\infty[$, $m \in \mathbb{N}_+$, and $\overline{D}$ being the closure of $D$. The reason for considering the closure is tied to PDE applications, where it is necessary to evaluate functions on the domain's boundary.

**Theorem 3.** *Let $\sigma$, $\mathcal{A}$ and $\mathcal{U}$ be set as above. Let $\mathcal{G}\colon \mathcal{A} \to \mathcal{U}$ be a continuous operator. Then for any compact set $K \subset \mathcal{A}$ and $\varepsilon > 0$ there exists a Bregman neural operator $\mathcal{N}_\theta\colon \mathcal{A} \to \mathcal{U}$ of the type (2) such that each component depends on a finite dimensional Bregman neural network and* FIX

$$\sup_{u \in K} \|\mathcal{G}(u) - \mathcal{N}_\theta(u)\|_{\mathcal{U}} \le \varepsilon.$$

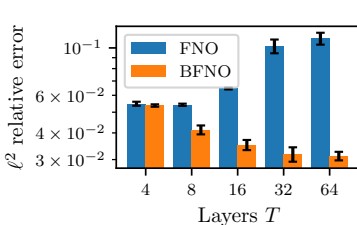 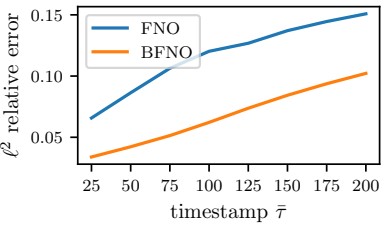

(a) Performance comparison

(b) Performance in autoregressive mode

Figure 4: Results on 1D Burgers (viscosity $\nu = 10^{-3}$)

*Here $\theta \in \mathbb{R}^p$ collects all the (finite number of) parameters of the finite dimensional Bregman neural networks defining the components in* (2).

This theorem is based on the fact that we were able to prove this same result for Bregman neural networks in finite dimensional spaces. See Appendix B.

## 5 NUMERICAL EXPERIMENTS

The primary objective of our numerical experiments is to evaluate and assess the added benefits of the Bregman variant of the simplest neural operator, namely Fourier Neural Operator (FNO) as it often serves as the building block for more sophisticated models. Additional models such as WNO (Tripura & Chakraborty, 2023) and its Bregman variant, F-FNO (Tran et al., 2023) and a ResNet-like variant of FNO are studied in the appendix.

NEW

### 5.1 EXPERIMENTAL SETTING

**Datasets.** We have selected a range of benchmark datasets resulting from the resolution of PDEs used both in the original FNO paper (Li et al., 2021a) and in the PDEBench suite (Takamoto et al., 2022), which is the top leading repository providing datasets commonly studied in physics-based machine learning. They represent various dynamics and complexities pertinent to physical modeling tasks. Hereafter, we consider initial value problems where the goal is to learn the mapping between the initial condition $a_i$ and the solution at some future time $u_i$ from $n = 10^4$ pairs $\{a_i, u_i\}_{i=1}^n$. A description of the experimental settings and the learning procedure is provided in Appendix C.

**Models.** We consider the FNO (Li et al., 2021a) and its Bregman variant (BFNO), described in Section 3.4. Note that, by design, both versions yield the same training time and memory usage. The lifting and projection layers, namely $\mathcal{P}$ and $\mathcal{Q}$ in (2), are convolutional layers with kernel size 1 and width 128. Note that, for BFNO, we add an activation operator after $\mathcal{P}$ to ensure that the conditions of Remark 4 (i) are met. Following the code of Li et al. (2021a), we use the ReLU activation for FNO while, for BFNO, we resort to an invertible approximation: SoftPlus with parameter $\beta = 10^3$ to make it almost indistinguishible from ReLU. Hereafter, we consider models made of $T \in \{4, 8, 16\}$ Fourier layers with a width 64 (resp. 32) and 16 (resp. 12) maximum number of Fourier modes for 1D (resp. 2D) problems. Note that two ablation studies in Appendices D.5 and D.6 reveal marginal improvements from adding batch normalization layers or replacing SoftPlus with ReLU.

NEW

### 5.2 RESULTS AND ANALYSIS

**Illustration and impact of the number of layers $T$.** We illustrate the behavior of the prediction error as the number of operator layers $T$ increases. To this end, we conducted an experiment using the Burgers' dataset with viscosity $\nu = 10^{-3}$, with results presented in Figure 4a. First, we observe that BFNO systematically yields lower prediction error, irrespectively of $T$. Second, the performance of FNO degrades starting from $T = 16$, while BFNO demonstrates better performance as $T$ increases until it reaches a plateau at $T = 64$. The same conclusion holds for other datasets and our Bregman variant of WNO, as illustrated in Appendix D.2. We believe that this interesting property is due to the added term of BFNO which helps in stabilizing the learning since BFNO layers reduce to the identity

NEW

Table 2: Relative error of FNO and BFNO models on benchmark PDEs.

| | 4 layers | | 8 layers | | 16 layers | |
|---|---|---|---|---|---|---|
| | FNO | BFNO | FNO | BFNO | FNO | BFNO |
| 1D Advection | $1.0 \pm 0.0\%$ | $\mathbf{0.7} \pm 0.0\%$ | $1.4 \pm 0.1\%$ | $\mathbf{0.6} \pm 0.1\%$ | $1.8 \pm 0.1\%$ | $\mathbf{0.6} \pm 0.1\%$ |
| 1D Burgers ($\nu = 10^{-1}$) | $0.5 \pm 0.0\%$ | $\mathbf{0.3} \pm 0.0\%$ | $0.7 \pm 0.0\%$ | $\mathbf{0.3} \pm 0.0\%$ | $0.9 \pm 0.0\%$ | $\mathbf{0.4} \pm 0.0\%$ |
| 1D Burgers ($\nu = 10^{-3}$) | $5.5 \pm 0.1\%$ | $\mathbf{5.4} \pm 0.1\%$ | $5.4 \pm 0.1\%$ | $\mathbf{4.1} \pm 0.2\%$ | $6.5 \pm 0.1\%$ | $\mathbf{3.5} \pm 0.2\%$ |
| 2D NS ($\nu = 10^{-3}$) | $4.6 \pm 0.1\%$ | $\mathbf{4.3} \pm 0.1\%$ | $4.1 \pm 0.1\%$ | $\mathbf{4.0} \pm 0.0\%$ | $\mathbf{3.9} \pm 0.1\%$ | $4.0 \pm 0.1\%$ |
| 2D NS ($\nu = 10^{-4}$) | $\mathbf{13.5} \pm 0.1\%$ | $13.7 \pm 0.1\%$ | $13.0 \pm 0.2\%$ | $\mathbf{12.6} \pm 0.1\%$ | $12.6 \pm 0.1\%$ | $\mathbf{12.2} \pm 0.1\%$ |
| 1D NS | $58.2 \pm 0.6\%$ | $\mathbf{57.0} \pm 0.6\%$ | $58.2 \pm 0.6\%$ | $\mathbf{56.8} \pm 0.8\%$ | $59.7 \pm 0.6\%$ | $\mathbf{56.5} \pm 0.6\%$ |
| 2D Darcy | $34.6 \pm 0.0\%$ | $\mathbf{33.4} \pm 0.2\%$ | $32.8 \pm 0.2\%$ | $\mathbf{31.5} \pm 0.4\%$ | $32.9 \pm 0.2\%$ | $\mathbf{30.0} \pm 0.5\%$ |

when all the weights are zero. In Figure 6a, we report one instance of an input-output pair and the best predicted output by FNO and BFNO, showing that BFNO better predicts the sharp edges. An analysis of the weight probability density distribution is provided in Appendix D.7.

**Learning the solution map.** As previously mentioned, we consider the problem of learning the    FIX
mapping between the initial condition and the solution of a PDE at some future time. In Table 2, we compare the prediction performance, in terms of $\ell^2$ relative error, between FNO and BFNO for $T = \{4, 8, 16\}$ layers across different PDEs of varying complexities. Results indicate that BFNO consistently yields better or comparable prediction performance. Additionally, the behavior observed with the Burgers' PDE, where the performance improves or stabilizes without degrading as $T$ increases, also holds for other PDEs. In contrast, FNO may suffer from a degradation of performance. The reader may refer to Appendix D.2 for an analysis up to 64 layers, where we demonstrate that the favorable behavior of the Bregman variant extends not only to FNOs but also to WNOs. In addition, a detailed version of Table 2 is provided in Appendix D.3, where the prediction performance is also analyzed both in frequency bands and on the boundary of the domain, leading to similar conclusions. This behavior underscores BFNO's ability to circumvent challenges commonly encountered when training deep models, a capability further demonstrated through additional experiments detailed in Appendix D.4, which highlight its advantages over F-FNO and a ResNet-like FNO.

**Learning the time-step evolution map.** We now consider the problem of learning the mapping    NEW
between the solution at some time $t$ and the solution at $t + 25$. Then we pose our model in an autoregressive mode, where the output is fed again to the input of the model, repeating it 8 times. Results provided in Figure 4b show that BFNO actually benefits from better prediction at each horizon.

## 6    CONCLUSION

In summary, our contributions are twofold: we have provided a novel theoretical framework that broadens the understanding of neural operators through the lens of a Bregman regularized optimization problem, and we have introduced Bregman neural operators that achieve enhanced performance as their depth increases. As part of our theoretical advancements, we have also established universal approximation results for Bregman neural architectures with sigmoidal-type activation functions. However, it must be acknowledged that a gap exists between this result and common practices, which predominantly rely on ReLU-like activations, as in our work, opening the door to new theoretical developments. Beyond the unifying aspect of our framework and its ability to design novel neural architectures, our framework also paves the way to use the rich body of literature on    NEW
monotone operators to study neural operators. In the context of neural networks, an example of fruitful application of the latter is given in Combettes & Pesquet (2020a) where the authors provide asymptotic properties of neural networks (as the number of layers tends to infinity). One can also consider the work in Combettes & Pesquet (2020b) where the authors yield quantitative insights into the stability properties of neural networks. As for our setting, we can guess that such results might be extended to Bregman neural networks/operators by leveraging the notion of so called D-firm operators studied in Bauschke et al. (2003), meaning operators that are firmly nonexpansive with respect to a Bregman divergence.

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

## A   ADDITIONAL TECHNICAL FACTS

We begin by introducing the necessary notations used throughout the paper.

**Notations.** Let $\mathcal{V}$ and $\mathcal{V}^*$ be two Banach spaces put in duality via the pairing $\langle \cdot, \cdot \rangle \colon \mathcal{V} \times \mathcal{V}^* \to \mathbb{R}$. If $\Phi \colon \mathcal{V} \to \, ]-\infty, +\infty]$, we denote by $\mathrm{dom}\,\Phi = \{v \in \mathcal{V} \,|\, \Phi(v) < +\infty\}$ its *effective domain*. For every proper convex function $\Phi \colon \mathcal{V} \to \, ]-\infty, +\infty]$, we set its subdifferential

$$\partial \Phi(v) = \{v^* \in \mathcal{V}^* \,|\, \text{for all } u \in \mathcal{V}, \ \Phi(u) \geq \Phi(v) + \langle u - v, v^* \rangle\},$$

if $v \in \mathrm{dom}\,\Phi$, and $\partial\Phi(v) = \varnothing$, otherwise. We set $\mathrm{dom}\,\partial\Phi = \{v \in \mathrm{dom}\,\Phi \,|\, \partial\Phi(v) \neq \varnothing\}$ and the *range* $\mathrm{ran}\,\partial\Phi = \{v^* \in \mathcal{V}^* \,|\, \exists v \in \mathcal{V} \text{ s.t. } v^* \in \partial\Phi(v)\}$. When $\partial\Phi(v)$ is a singleton, we denote by $\widetilde{\nabla}\Phi$ its unique element. If $\Phi \colon \mathcal{V} \to \, ]-\infty, +\infty]$, its *Fenchel conjugate* is the function $\Phi^* \colon \mathcal{V}^* \to \, ]-\infty, +\infty]$ such that $\Phi^*(v^*) = \sup_{v \in \mathcal{V}} \langle v, v^* \rangle - \Phi(v)$. We denote by $\Gamma_0(\mathcal{V})$ the set of proper convex and lower-semicontinuous functions on $\mathcal{V}$. The Fenchel-Moreau theorem ensures that $\Phi \in \Gamma_0(\mathcal{V}) \Rightarrow \Phi^* \in \Gamma_0(\mathcal{V}^*)$. We denote by $\langle \cdot, \cdot \rangle$ and $|\cdot|$ the Euclidean scalar product and norm in $\mathbb{R}^n$. If $D \subset \mathbb{R}^d$ is a nonempty bounded Borel set and $p \in [1, +\infty]$, we denote by $L^p(D, \mathbb{R}^n)$ the Lebesgue space of $p$-integrable functions (essentially bounded functions, if $p = +\infty$) from $D$ to $\mathbb{R}^n$.

NEW

MOVE

### A.1   CONSIDERATIONS FOR LEGENDRE FUNCTION AND BREGMAN PROXIMAL OPERATORS

At the core of our framework, lies the connection between activation operators and Bregman proximity operators whose definition involves the Bregman divergence itself defined from a Legendre function $\Phi \in \Gamma_0(\mathcal{V})$. The latter acts on Lebesgue function space $\mathcal{V} = L^p(D, \mathbb{R}^n)$ and can be built from an elementary legendre function $\phi \in \Gamma_0(\mathbb{R}^n)$ through the convex integral functional described in Fact 1. We provide below several considerations.

NEW

**Remark 6.** *One can prove that $\phi$ is Legendre if and only if $\phi^*$ is Legendre. Moreover, if $\phi$ is Legendre, then $\phi$ and $\phi^*$ are differentiable on $\mathrm{int}(\mathrm{dom}\,\phi)$ and $\mathrm{int}(\mathrm{dom}\,\phi^*)$ respectively and*

$$\nabla\phi \colon \mathrm{int}(\mathrm{dom}\,\phi) \to \mathrm{int}(\mathrm{dom}\,\phi^*) \quad \text{and} \quad \nabla\phi^* \colon \mathrm{int}(\mathrm{dom}\,\phi^*) \to \mathrm{int}(\mathrm{dom}\,\phi)$$

*are bijective and inverse of each other.*

MOVE

**Remark 7.** *In Fact 1, suppose that $p = 1$ and $\mathrm{dom}\,\phi^* = \mathbb{R}^n$. Then $\mathrm{ran}\,\partial\Phi = \mathcal{V}^*$. Indeed, we note that $\nabla\phi \colon \mathrm{int}(\mathrm{dom}\,\phi) \to \mathbb{R}^n$ is a continuous bijection with inverse $\nabla\phi^*$, which is also continuous. Therefore if we let $u \in \mathcal{V}^* = L^\infty(D, \mathbb{R}^n)$ and set $v = (\nabla\phi^*) \circ u$, since $u$ is essentially bounded, we have that $v$ is essentially bounded too, and hence integrable. In the end $v \in L^1(D, \mathbb{R}^n)$ and $u = (\nabla\phi) \circ v \in \partial\Phi(v)$.*

Definition 3 of Bregman proximity operators in general Banach spaces requires that $\mathrm{ran}\,\partial(\Phi + g)$ is the full dual space. The following result gives a simple situation in which such condition is satisfied.

MOVE

**Proposition 4.** *Let $\phi \in \Gamma_0(\mathbb{R}^n)$ be a Legendre function, let $p \in [1, +\infty[$, and suppose that $\phi$ is $p$-uniformly convex with constant $c > 0$, meaning that*

$$\forall y, y' \in \mathbb{R}^n, \forall \lambda \in \, ]0, 1[ \, : \, \phi((1-\lambda)y + \lambda y') + \lambda(1-\lambda)\frac{c}{p}|y - y'|^p \leq (1-\lambda)\phi(y) + \lambda\phi(y'). \tag{12}$$

*Let $\mathcal{V} = L^p(D, \mathbb{R}^n)$. Then the integral functional $\Phi \colon \mathcal{V} \to \, ]-\infty, +\infty]$ defined as in Fact 1 is $p$-uniformly convex with respect to the norm $\|\cdot\|_p$. Moreover, for every $g \in \Gamma_0(\mathcal{V})$ such that $\mathrm{dom}\,\Phi \cap \mathrm{dom}\,g \neq \varnothing$, we have $\mathrm{dom}(\Phi + g)^* = \mathcal{V}^*$ and $(\Phi + g)^*$ is Fréchet differentiable on $\mathcal{V}^*$. Thus $\mathcal{V}^* = \mathrm{dom}\,\partial(\Phi + g) = \mathrm{ran}\,\partial(\Phi + g)$.*

*Proof.* It follows by integrating (12). The second part follows by Zalinescu (2002, Theorem 3.5.10), considering that $\Phi + g$ is also $p$-uniformly continuous. $\qquad\square$

### A.2   LINK BETWEEN ACTIVATION FUNCTION AND PROXIMITY OPERATOR

As demonstrated in the work of Combettes & Pesquet (2020a), many activation functions $\sigma$ can be expressed as proximity operators $\mathrm{prox}_g = \mathrm{argmin}_{t \in \mathbb{R}} g(t) + \frac{1}{2}(\cdot - t)^2$ for some appropriate convex function $g$. The simplest case is that of the ReLu activation function, recalled below.

**Example 1** (ReLu). *The rectified linear unit function $\sigma \colon t \in \mathbb{R} \mapsto \max(t, 0) \in \mathbb{R}$ can be expressed as the proximity operator $\mathrm{prox}_g$ of $g = \imath_{[0,+\infty[}$. Henceforth, $\mathrm{prox}_g$ reduces to the projection onto the positive orthant.*

We also provide a novel characterization of SoftPlus.

**Example 2** (SoftPlus). *Given $\beta > 0$, the SoftPlus activation function, i.e., $\sigma \colon t \mapsto \mathrm{SoftPlus}_\beta(t) \triangleq (1/\beta) \log(\exp(\beta t) + 1)$, is the proximity operator of*

$$g \colon t \in \mathbb{R}_{>0} \mapsto \frac{1}{\beta^2} \mathrm{Li}_2(\mathrm{e}^{-\beta t}) \in \mathbb{R}_{>0}, \tag{13}$$

*where $\mathrm{Li}_2$ is the dilogarithm function defined as $\mathrm{Li}_2 \colon t \mapsto -\int_0^t \frac{\log(1-u)}{u} \mathrm{d}u$.*

*Proof.* For every $s \in \mathbb{R}$, $\mathrm{prox}_g(s) = \mathrm{argmin}_{t \in \mathbb{R}} \{h(t) \triangleq g(t) + (1/2)(s - t)^2\}$ with $h(t) = (1/\beta^2)\mathrm{Li}_2(\mathrm{e}^{-\beta t}) + (1/2)(s - t)^2 = \psi(t) - st + (1/2)s^2$ where we introduced $\psi(t) = (1/\beta^2)\left(\mathrm{Li}_2(\mathrm{e}^{-\beta t}) + (1/2)\log(\mathrm{e}^{-\beta t})^2\right) = (1/\beta^2)\int^{\mathrm{e}^{-\beta t}} \log(r/(1-r))/r \mathrm{d}r$. The latter can be written as $\psi(t) = (1/\beta)\int^t \log(\mathrm{e}^{\beta r} - 1)\mathrm{d}r$ up to a constant. Finally, since $h$ is strongly convex, the minimum is attained for $t$ such that $h'(t) = 0$, which yields $\log(\mathrm{e}^{\beta t} - 1) = \beta s \Leftrightarrow t = \sigma(s)$, thus ending the proof. $\square$

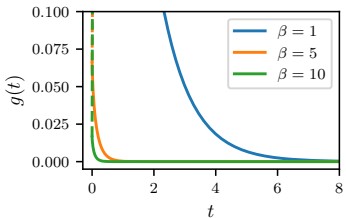 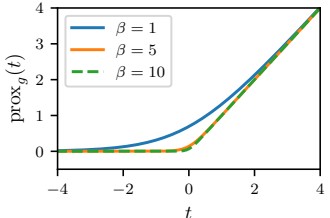

(a) Representation of $t \mapsto g(t) = \frac{1}{\beta^2} \mathrm{Li}_2(\mathrm{e}^{-\beta t})$     (b) Representation of $\mathrm{prox}_g(t) = \mathrm{SoftPlus}_\beta(t)$

Figure 5: **Illustration of SoftPlus as a proximity operator.**

We present an illustration of the convex function $g$ defined in Eq. 13 in Figure 5a. Intuitively, it serves as a smooth surrogate for the indicator function of the positive orthant $\imath_{[0,+\infty[}$. A larger value of $\beta > 0$ leads to a closer approximation. This aligns with the representation of SoftPlus as the proximity operator of $g$ from Eq. 13, depicted in Fig. 5b where a larger $\beta$ makes SoftPlus closer to ReLU.

# B  APPROXIMATION RESULTS FOR BREGMAN NEURAL NETWORKS AND OPERATORS

## B.1  BREGMAN NEURAL NETWORKS

We consider first shallow Bregman neural networks for finite dimensional spaces. Let $\sigma \colon \mathbb{R} \to I$ be a homeomorphism, where $I$ is an open interval in $\mathbb{R}$. We $d \in \mathbb{N}_+$ and set

$$\mathrm{BN}_2(\sigma; I^d) = \mathrm{span}\{\sigma(\sigma^{-1}(m^\top x) + w^\top x + b) \mid m \in \Delta^{d-1}, w \in \mathbb{R}^d, b \in \mathbb{R}\}. \tag{14}$$

**Remark 8.** *Since $m$ belongs to the standard simplex $\Delta^{d-1}$, $m^\top x$ is a convex combination of elements of $I$ and so it is an element of $I$. Thus, since $\sigma^{-1} \colon I \to \mathbb{R}$, the functions in $\mathrm{BN}_2(\sigma; I^d)$ are well-defined from $I^d \to \mathbb{R}$.*

The following result follows from an adaptation of the argument in Cybenko (1989) to our different architecture (14).

**Theorem 5.** *Suppose that $\sigma$ is sigmoidal, meaning that $\lim_{t \to -\infty} \sigma(t) = 0$ and $\lim_{t \to +\infty} \sigma(t) = 1$. Then, the space $\mathrm{BN}_2(\sigma; I^d)$ is dense in $\mathcal{C}(I^d, \mathbb{R})$ with respect to the topology of uniform convergence on compact sets.*

*Proof.* Let $K \subset I^d$ be a compact set. We prove that the trace space $\mathsf{BN}_2(\sigma; I^d)_{|K}$ is dense in $\mathcal{C}(K, \mathbb{R})$. To that purpose, we rely on the following general fact concerning dense sets in Banach space (see, e.g., Brezis (2011)). Let $\mathcal{B}$ be a Banach space, let $\mathcal{A} \subset \mathcal{B}$. Then the following propositions are equivalent.

- span$\mathcal{A}$ is dense in $\mathcal{B}$

- $\mathcal{A}^\perp = \{u^* \in \mathcal{B}^* \,|\, \forall u \in \mathcal{A} : \langle u, u^* \rangle = 0\} = \{0\}$.

- $\forall u^* \in \mathcal{B}^*, (\forall u \in \mathcal{A} : \langle u, u^* \rangle = 0) \Rightarrow u^* = 0$.

This implies that for our purpose we can equivalently prove that

$$\forall \mu \in \mathcal{M}(K) : \left( \forall f \in \mathsf{BN}_2(\sigma; I^d) : \int_K f\mu = 0 \right) \Rightarrow \mu = 0,$$

where $\mathcal{M}(K)$ is the space of signed finite Radon measures on $K$ (the dual of $\mathcal{C}(K)$). Thus, let $\mu$ be a signed measure on $K$ and suppose that

$$\forall f \in \mathsf{BN}_2(\sigma; I^d) : \int_K f d\mu = 0. \tag{15}$$

Fix $w \in \mathbb{R}^d, m \in \Delta^{d-1}$, and $b \in \mathbb{R}$. Define, for every $\lambda > 0$ and $c \in \mathbb{R}$

$$\sigma_{\lambda,c} : I \to \mathbb{R}, \quad x \mapsto \sigma(\sigma^{-1}(m^\top x) + \lambda(w^\top x + b) + c).$$

It is clear that $\sigma_{\lambda,c} \in \mathsf{BN}_2(\sigma; I^d)$. Moreover,

$$\lim_{\lambda \to +\infty} \sigma_{\lambda,c}(x) = \left\{ \begin{array}{ll} 1 & \text{if } w^\top x + b > 0 \\ 0 & \text{if } w^\top x + b < 0 \\ \sigma(\sigma^{-1}(m^\top x) + c) & \text{if } w^\top x + b = 0. \end{array} \right\} := \gamma(x).$$

Define the sets

$$\Pi^+_{w,b} = \{x \in K \,|\, w^\top x + b > 0\}, \quad \Pi^-_{w,b} = \{x \in K \,|\, w^\top x + b < 0\}, \quad \Pi_{w,b} = \{x \in K \,|\, w^\top x + b = 0\}.$$

They are intersections of half-spaces and hyperplanes with $K$. So,

$$\gamma(x) = \chi_{\Pi^+_{w,b}}(x) + \sigma(\sigma^{-1}(m^\top x) + c)\chi_{\Pi_{w,b}}(x),$$

where $\chi_A$ is the characteristic functions of the set $A \subset I^d$. Since $\sigma$ is bounded we can apply the Lebesgue's dominated convergence theorem and get

$$\lim_{\lambda \to +\infty} \underbrace{\int_K \sigma_{\lambda,c} d\mu}_{=0} = \int_K \gamma d\mu = \mu(\Pi^+_{w,b}) + \int_{\Pi_{w,b}} \sigma(\sigma^{-1}(m^\top x) + c) d\mu(x).$$

Note that the integral on the left is zero by the hypothesis (15). In this way we proved that

$$\forall m \in \Delta^{d-1}, \forall w \in \mathbb{R}^d, \forall b, \forall c \in \mathbb{R} : \quad \mu(\Pi^+_{w,b}) + \int_{\Pi_{w,b}} \sigma(\sigma^{-1}(m^\top x) + c) d\mu(x) = 0. \tag{16}$$

Now observe that (16) implies

$$\left| \mu(\Pi^+_{w,b}) \right| = \left| \int_{\Pi_{w,b}} \sigma(\sigma^{-1}(m^\top x) + c) d\mu(x) \right| \le \int_{\Pi_{w,b}} |\sigma(\sigma^{-1}(m^\top x) + c)| \, d|\mu|(x) \to 0 \text{ as } c \to -\infty,$$

since $|\sigma(\sigma^{-1}(m^\top x) + c)| \to 0$ as $c \to -\infty$ (pointwise), where $|\mu|$ is the total variation of $\mu$. Therefore, $\mu(\Pi^+_{w,b}) = 0$. Then (16) yields

$$\forall c \in \mathbb{R} : \quad \int_{\Pi_{w,b}} \sigma(\sigma^{-1}(m^\top x) + c) d\mu(x) = 0.$$

Moreover, by assumption $\sigma(\sigma^{-1}(m^\top x) + c) \to 1$ as $c \to +\infty$ (pointwise) and hence, again by Lebesgue's dominated convergence theorem,

$$\lim_{c \to +\infty} \underbrace{\int_{\Pi_{w,b}} \sigma(\sigma^{-1}(m^\top x) + c)d\mu(x)}_{=0} = \int_{\Pi_{w,b}} 1 d\mu = \mu(\Pi_{w,b}),$$

which yields $\mu(\Pi_{w,b}) = 0$. In the end we proved that the measure $\mu$ is zero on all the sets of type

$$\Pi_{w,b} \quad \text{and} \quad \Pi_{w,b}^+.$$

Now the proof continues as in Cybenko (1989, Lemma 1), and we can conclude that $\mu = 0$. $\qquad\square$

Now we address the vectorial case. We set

$$\mathsf{BN}_2(\sigma; I^d, \mathbb{R}^k) := \left\{ Q\sigma(\sigma^{-1}(Mx) + Wx + b) \,\middle|\, \begin{array}{l} r \in \mathbb{N}_+, Q \in \mathbb{R}^{k\times r}, W, M \in \mathbb{R}^{r\times d}, \\ \text{with } M \text{ right stochastic, and } b \in \mathbb{R}^r \end{array} \right\},$$

where $\sigma$ and $\sigma^{-1}$ are applied component-wise.

**Corollary 6.** *We have that*

$$\mathsf{BN}_2(\sigma; I^d, \mathbb{R}^k) = (\mathsf{BN}_2(\sigma; I^d))^k := \underbrace{\mathsf{BN}_2(\sigma; I^d) \times \cdots \times \mathsf{BN}_2(\sigma; I^d)}_{k \ times} \qquad (17)$$

*and it is dense in $\mathcal{C}(I^d, \mathbb{R}^k)$, in the topology of uniform convergence on compact sets.*

*Proof.* In view of Theorem 5, it is clear that $(\mathsf{BN}_2(\sigma; I^d))^k$ is dense in $\mathcal{C}(I^d, \mathbb{R})^k \cong \mathcal{C}(I^d, \mathbb{R}^k)$ in the topology of uniform convergence on compact sets. Let's prove equality (17). The inclusion $\mathsf{BN}_2(\sigma; I^d, \mathbb{R}^k) \subset (\mathsf{BN}_2(\sigma; I^d))^k$ is immediate. Let $f: I^d \to \mathbb{R}^k$ with components $f_j \in \mathsf{BN}_2(\sigma; I^d)$, $j = 1, \ldots, k$. Then, there exists $r \in \mathbb{N}_+$, and for each $j \in \{1, \ldots, k\}$, $q_j \in \mathbb{R}^r$, $W_j \in \mathbb{R}^{r\times d}$, $b_j \in \mathbb{R}^r$, and $M_j \in \mathbb{R}^{r\times d}$ right stochastic matrix (the rows are positive and sum one), such that

$$f_j(x) = q_j^\top \sigma(\sigma^{-1}(M_j x) + W_j x + b_j).$$

Then considering the block matrices

$$M = \begin{bmatrix} M_1 \\ \vdots \\ M_k \end{bmatrix} \in \mathbb{R}^{kr \times d}, \quad W = \begin{bmatrix} W_1 \\ \vdots \\ W_k \end{bmatrix} \in \mathbb{R}^{kr \times d}, \quad b = \begin{bmatrix} b_1 \\ \vdots \\ b_k \end{bmatrix} \in \mathbb{R}^{kr}, \quad Q = \begin{bmatrix} q_1^\top & 0 & \cdots & 0 \\ 0 & q_2^\top & \cdots & 0 \\ \vdots & \vdots & \ddots & \vdots \\ 0 & 0 & \cdots & q_k^\top \end{bmatrix} \in \mathbb{R}^{k \times kr},$$

we have

$$f(x) = Q\sigma(\sigma^{-1}(Mx) + Wx + b),$$

and hence $f \in \mathsf{BN}_2(\sigma; I^d, \mathbb{R}^k)$. The statement follows. $\qquad\square$

A general deep Bregman neural network with $T$ layers is defined as follows

$$\mathsf{BN}_T(\sigma; I^d, \mathbb{R}^k) = \{W_T \circ L_{T-1} \circ \cdots \circ L_1\},$$

where, for every $t = 1, \ldots, T - 1$,

$$L_t: I^{n_{t-1}} \to I^{n_t}, \quad x \mapsto \sigma(\sigma^{-1}(M_t x) + W_t x + b_t), \qquad (18)$$

with $W_t \in \mathbb{R}^{n_t \times n_{t-1}}, b_t \in \mathbb{R}^{n_t}$ and $M_t \in \mathbb{R}^{n_t \times n_{t-1}}$ right stochastic, for $t = 1, \ldots, T - 1$, with $n_0 = n$ and $W_T \in \mathbb{R}^{k \times n_{T-1}}$. Note that also the dimensions $n_1, \ldots, n_{T-1}$ can be chosen freely. Clearly for a deep network with $T > 2$, if we take, for every $t = 2, \ldots, T - 1$, $n_t = n_1$, $W_t = 0$, $b_t = 0$, and $M_t$ equals to the identity, then the layers $L_t$ with $t = 2, \ldots, T - 1$ act as the identity operator and hence

$$\mathsf{BN}_2(\sigma; I^d, \mathbb{R}^k) \subset \mathsf{BN}_T(\sigma; I^d, \mathbb{R}^k).$$

Therefore, $\mathsf{BN}_T(\sigma; I^d, \mathbb{R}^k)$ is dense in $\mathcal{C}(I^d, \mathbb{R}^k)$ for the topology of uniform convergence on compact sets.

**Remark 9.** *Often in applications it is desirable to have functions defined on the entire space $\mathbb{R}^d$. In this case one can simply precompose the functions in $\mathsf{BN}_T(\sigma; I^d, \mathbb{R}^k)$ by the homeomorphism*

$$x \in \mathbb{R}^d \to \sigma(x) \in I^d$$

*obtaining a dense set in $\mathcal{C}(\mathbb{R}^d, \mathbb{R}^k)$ (for any $T \geq 2$). Such space is then denoted by $\mathsf{BN}_T(\sigma; \mathbb{R}^d, \mathbb{R}^k)$.*

Let $D \subset \mathbb{R}^d$ be any nonempty bounded open set. If $\mathcal{F}(\mathbb{R}^d)$ is any class of real functions from $\mathbb{R}^d$ to $\mathbb{R}$ we denote by $\mathcal{F}_{|\overline{D}}$ the set of restrictions to $\overline{D}$ of the functions in $\mathcal{F}(\mathbb{R}^d)$. In the following according to Remark 9 we put

$$\mathsf{BN}_T(\sigma; \mathbb{R}^d, \mathbb{R}^k) = \big\{ W_T \circ L_{T-1} \circ \cdots \circ L_1 \circ \sigma \big\}, \tag{19}$$

which is a dense space in $\mathcal{C}(\mathbb{R}^d, \mathbb{R}^k)$ with respect to the topology of uniform convergence on compact sets.

**Lemma 7.** *Suppose that $\sigma$ is a sigmoidal activation function as in Theorem 5. Let $p \in [1, +\infty[$. Then $\mathsf{BN}_T(\sigma; \mathbb{R}^d, \mathbb{R}^k)_{|\overline{D}}$ is dense in $L^p(D, \mathbb{R}^k)$ (in the norm $\|\cdot\|_p$).*

*Proof.* It is well known that $\mathcal{C}_c(D, \mathbb{R}^k)$ is dense in $L^p(D, \mathbb{R}^k)$ and hence $\mathcal{C}(\mathbb{R}^n, \mathbb{R}^k)_{|\overline{D}}$ is dense in $L^p(D, \mathbb{R}^k)$ (in the norm $\|\cdot\|_p$). Moreover, $\mathsf{BN}_T(\sigma; \mathbb{R}^n, \mathbb{R}^k)_{|\overline{D}}$ is dense in $\mathcal{C}(\mathbb{R}^d, \mathbb{R}^k)_{|\overline{D}}$ (in the norm $\|\cdot\|_\infty$). On the other hand

$$\forall f \in \mathcal{C}(\mathbb{R}^d, \mathbb{R}^k)_{|\overline{D}}: \quad \|f\|_p = \Big( \int_D |f|^p dx \Big)^{1/p} \leq \|f\|_\infty |D|^{1/p}.$$

Thus, if $f \in L^p(D, \mathbb{R}^k)$ and $\varepsilon > 0$,

$$\exists\, g \in \mathcal{C}(\mathbb{R}^d, \mathbb{R}^k)_{\overline{D}} \text{ s.t. } \|f - g\|_p \leq \frac{\varepsilon}{2}$$

$$\exists\, h \in \mathsf{BN}_T(\sigma; \mathbb{R}^d, \mathbb{R}^k)_{|\overline{D}} \text{ s.t. } \|g - h\|_\infty \leq \frac{\varepsilon}{2|D|^{1/p}} \ \Rightarrow \ \|g - h\|_p \leq \frac{\varepsilon}{2}$$

and hence $\|f - h\|_p \leq \varepsilon$. $\qquad\qquad\qquad\qquad\qquad\qquad\qquad\qquad\qquad\qquad\square$

**Remark 10.** *It is sometimes required that neural networks, of any depth, include constant functions. Standard feed-forward neural networks have the form*

$$(W_T \ \cdot + b_T) \circ \sigma(W_{T-1} \ \cdot\ + b_{T-1}) \circ \cdots \circ \sigma(W_1 \ \cdot\ + b_1),$$

*so it is clear that they include constant functions (just take $W_T = 0$). However, for Bregman neural networks as defined in (19)-(18) this is not clear. An immediate modification to achieve this goal is to explicitly add a constant $b_T$ in the last layer. Another possibility is to lift the input space by one dimension, precomposing the neural network with a (free) linear embedding. In particular, if we consider the canonical embedding*

$$J\colon \mathbb{R}^d \to \mathbb{R}^{d+1}\colon x \mapsto \begin{bmatrix} x \\ 0 \end{bmatrix},$$

*and define the following matrices*

$$\tilde{W}_t = \begin{bmatrix} W_t & 0 \\ 0 & 1 \end{bmatrix}, \quad \tilde{M}_t = \begin{bmatrix} M_t & 0 \\ 0 & 1 \end{bmatrix}, \quad \tilde{b}_t = \begin{bmatrix} b_t \\ -\sigma(0) \end{bmatrix}, \quad \text{(for } t<T) \quad \tilde{W}_T = [W_T \quad b_T/\sigma(0)],$$

*then, for $t = 1, \ldots, T-1$, according to (18), we have*

$$\forall\, y \in I^{n_{t-1}}\colon \tilde{L}_t \begin{bmatrix} y \\ \sigma(0) \end{bmatrix} = \sigma\Big( \sigma^{-1}\Big( \tilde{M}_t \begin{bmatrix} y \\ \sigma(0) \end{bmatrix} \Big) + \tilde{W}_t \begin{bmatrix} y \\ \sigma(0) \end{bmatrix} + \tilde{b}_t \Big) = \begin{bmatrix} L_t y \\ \sigma(0) \end{bmatrix}$$

*and hence*

$$\tilde{W}_T \circ \tilde{L}_{T-1} \circ \cdots \circ \tilde{L}_1 \circ \sigma \circ J = W_T \circ L_{T-1} \circ \cdots \circ L_1 \circ \sigma + b_T.$$

## B.2 BREGMAN NEURAL OPERATORS

Now we start addressing the proof of Theorem 3. We will rely on the work of Kovachki et al. (2023), from which, for the sake of reader's convenience, we report the following facts.

**Fact 2** (Lemma 28 and 30 in Kovachki et al. (2023)). *Let $D \subset \mathbb{R}^d$ be a bounded set and let $L \in (W^{m,p}(D))^*$, for some $m \geq 0$ and $1 \leq p < +\infty$, or $L \in (\mathcal{C}(D))^*$. Then, for any closed and bounded set $K \subset \mathcal{A}$ and $\varepsilon > 0$, there exists a function $\kappa \in \mathcal{C}_c^\infty(D)$ such that*

$$\sup_{v \in K} \left| L(v) - \int_D \kappa(x)v(x)dx \right| < \varepsilon.$$

**Fact 3** (Lemma 22 and 26 in Kovachki et al. (2023)). *Let $D \subset \mathbb{R}^d$ be a bounded set and let $\mathcal{A}$ and $\mathcal{U}$ be any one of the Banach spaces $\mathcal{C}(\overline{D})$ or $W^{m,p}(D)$, with $m \geq 0$ and $1 \leq p < +\infty$. Let $\mathcal{G} \colon \mathcal{A} \to \mathcal{U}$ be a continuous operator, $K \subset \mathcal{A}$ be a compact set and $\varepsilon > 0$. Then there exist $J, J' \in \mathbb{N}$ and*

$$R \colon \mathcal{A} \to \mathbb{R}^J, \quad f \colon \mathbb{R}^J \to \mathbb{R}^{J'}, \quad S \colon \mathbb{R}^{J'} \to \mathcal{U},$$

*with $R$ and $S$ linear continuous and $f$ continuous, such that*

$$\sup_{v \in K} \|\mathcal{G}(v) - (S \circ f \circ R)(v)\| < \varepsilon.$$

In the following we set $D \subset \mathbb{R}^d$ be a bounded set and

$$\mathcal{A}(D, \mathbb{R}^{n_0}) = W^{m,p}(D, \mathbb{R}^{n_0}) \quad \text{or} \quad \mathcal{A}(D, \mathbb{R}^{n_0}) = \mathcal{C}(D, \mathbb{R}^{n_0}),$$

where the integer $m \geq 0$ and $p \in [1, +\infty[$. Moreover we will assume that (by possibly changing the definition slightly) Bregman neural networks include constant functions (recall Remark 10). Because of the density result given in the previous section, we can essentially follow the same line of arguments in Kovachki et al. (2023), but we need to take special care of the different structure of Bregman neural network/operators (in particular in Lemma 10).

**Lemma 8.** *Let $L \in \mathcal{A}^*$ and $K \subset \mathcal{A}$ be a compact set. Then there exists $h \in \mathsf{BN}_2(\sigma; \mathbb{R}^d, \mathbb{R}^{n_0})_{|D}$ such that*

$$\sup_{v \in K} \left| L(v) - \int_D \langle h(x), v(x) \rangle \, dx \right| < \varepsilon.$$

*Proof.* The space $\mathcal{A}$ is (isomorphic to) a product space, meaning $\mathcal{A} = \prod_{i=1}^{n_0} \mathcal{A}_i$, where $\mathcal{A}_i$ is a space of real valued functions on $D$. Set $K_i = \mathrm{pr}_i(K)$, which is a compact set of $\mathcal{A}_i$, so that $K \subset \prod_{i=1}^{n_0} K_i$. Then $L \colon \mathcal{A} \to \mathbb{R}$ can be written as $Lv = \sum_{i=1}^{n_0} L_i v_i$ with $L_i \colon \mathcal{A}_i \to \mathbb{R}$. By Fact 2, for every $i = 1, \ldots, n_0$, there exists $\kappa_i \in \mathcal{C}_c(D)$ such that

$$\sup_{v_i \in K_i} \left| L_i v_i - \int_D \kappa_i v_i \, dx \right| < \frac{\varepsilon}{2n_0}.$$

Let $\kappa \in \mathcal{C}_c(D, \mathbb{R}^{n_0})$ with components $\kappa_i \in \mathcal{C}_c(D)$. Then

$$\left| Lv - \int_D \langle \kappa(x), v(x) \rangle \, dx \right| = \left| \sum_{i=1}^{n_0} L_i v_i - \sum_{i=1}^{n_0} \int_D \kappa_i v_i \, dx \right| \leq \sum_{i=1}^{n_0} |L_i v_i - \int_D \kappa_i v_i \, dx| < \frac{\varepsilon}{2}.$$

Since $\mathcal{A} \subset L^1(D, \mathbb{R}^{n_0})$ we set $\gamma = \sup_{v \in K} \|v\|_1 < +\infty$. Moreover, since Bregman shallow neural networks are dense in the space of continuous functions (Remark 9), there exists $h \in \mathsf{BN}_2(\sigma; \mathbb{R}^d, \mathbb{R}^{n_0})_{|\overline{D}}$ such that $\|h - \kappa\|_\infty \leq \varepsilon/(2\gamma)$ and hence, for every $v \in K$,

$$\left| \int_D \langle \kappa, v \rangle \, dx - \int_D \langle h, v \rangle \, dx \right| = \left| \int_D \langle \kappa - h, v \rangle \, dx \right| \leq \int_D |\kappa(x) - h(x)||v(x)| \, dx \leq \|\kappa - h\|_\infty \|u\|_1 < \frac{\varepsilon}{2}.$$

Therefore,

$$\left| Lv - \int_D \langle h, v \rangle \, dx \right| \leq \left| Lv - \int_D \langle \kappa, v \rangle \, dx \right| + \left| \int_D \langle \kappa, v \rangle \, dx - \int_D \langle h, v \rangle \, dx \right| < \varepsilon$$

and the statement follows. $\qquad\square$

**Lemma 9.** *Let $R\colon \mathcal{A} \to \mathbb{R}^J$ be a linear continuous operator, $K \subset \mathcal{A}$ a compact set and $\varepsilon > 0$. Then there exists a linear continuous operator $R^{\mathsf{BN}}\colon \mathcal{A} \to \mathbb{R}^J$ acting as*

$$v \mapsto R^{\mathsf{BN}}v = \int_D h(y)v(y)\,dy,$$

*where $h \in \mathsf{BN}_2(\sigma; \mathbb{R}^d, \mathbb{R}^{J \times n_0})_{|\overline{D}}$, such that*

$$\sup_{v \in K} |Rv - R^{\mathsf{BN}}v| < \varepsilon.$$

*Proof.* Consider the components $R_j\colon \mathcal{A} \to \mathbb{R}$, $j = 1, \dots, J$. Then $R_j \in \mathcal{A}^*$, and by Lemma 8

$$\exists h_j \in \mathsf{BN}_2(\sigma; \mathbb{R}^d, \mathbb{R}^{n_0})_{|D} \quad \text{s.t.} \quad \sup_{v \in K} \left| R_j v - \int_D \langle h_j(x), v(x) \rangle \, dx \right| \le \frac{\varepsilon}{\sqrt{J}}.$$

Let $h\colon \mathbb{R}^d \to \mathbb{R}^{J \times n_0}$ with

$$h(x) = \begin{bmatrix} h_1(x)^\top \\ \vdots \\ h_J(x)^\top \end{bmatrix}.$$

Clearly $h \in \mathsf{BN}_2(\sigma; \mathbb{R}^d, \mathbb{R}^{J \times n_0})_{|D}$ and

$$\forall\, v \in K\colon \left| Rv - \int_D h(x)v(x)\,dx \right|^2 = \sum_{i=1}^J \left| R_j v - \int_D \langle h_j(x), v(x) \rangle \, dx \right|^2 < \varepsilon^2$$

and the statement follows. $\qquad\square$

**Remark 11.** *Both the linear continuous operators $R$ and $R^{\mathsf{BN}}$ in Lemma 9 can be canonically lifted to Lebesgue spaces as follows.*

$$\mathcal{R}\colon \mathcal{A} \to L^p(D, \mathbb{R}^J), \quad \mathcal{R}v = (Rv)\mathbb{1}_D$$
$$\mathcal{R}^{\mathsf{BN}}\colon \mathcal{A} \to L^p(D, \mathbb{R}^J), \quad \mathcal{R}^{\mathsf{BN}}v = (R^{\mathsf{BN}}v)\mathbb{1}_D,$$

*where $\mathbb{1}_D$ denotes the constant function $x \mapsto 1$ on $D$. Moreover $\mathcal{R}^{\mathsf{BN}}$ is actually an integral operator. Indeed if we define the kernel*

$$\kappa_h\colon D \times D \to \mathbb{R}^{J \times n_0}, \quad \kappa_h(x, y) = h(y)$$

*we have*

$$(\mathcal{R}^{\mathsf{BN}}v)(x) = R^{\mathsf{BN}}v = \int_D h(y)v(y)\,dy = \int_D \kappa_h(x, y)v(y)\,dy.$$

The following result is the analogue of Kovachki et al. (2023, Lemma 35) and establishes that a finite dimensional Bregman neural network can be canonically lifted in Lebesgue spaces. However, here we need to take care of the domain of the Bregman operator layers.

**Lemma 10.** *Let $f \in \mathsf{BN}_T(\sigma; \mathbb{R}^J, \mathbb{R}^{J'})$, $D \subset \mathbb{R}^d$ a nonempty open set and $p \in [1, +\infty]$. Then there exists a neural operator*

$$\mathcal{N}^{\mathsf{BN}}\colon L^p(D, \mathbb{R}^J) \to L^p(D, \mathbb{R}^{J'}), \quad \mathcal{N}^{\mathsf{BN}} = \mathcal{K}_T \circ \mathcal{L}_{T-1} \circ \cdots \circ \mathcal{L}_1 \circ \sigma,$$

*where, for every $t = 1, \dots, T - 1$,*

$$\mathcal{L}_t(v) = \sigma(\sigma^{-1}(\mathcal{M}_t v) + \mathcal{K}_t v + b_t)$$

*and such that the linear integral operators $\mathcal{M}_t$ and $\mathcal{K}_t$ and the functions $b_t$ are defined (parametrized) by finite dimensional Bregman shallow neural networks and*

$$\forall\, w \in \mathbb{R}^J\colon \mathcal{N}^{\mathsf{BN}}(w\mathbb{1}_D) = f(w)\mathbb{1}_D,$$

*where $\mathbb{1}_D$ denotes the constant function $x \mapsto 1$ on $D$.*

*Proof.* By definition

$$f = K_T \circ L_{T-1} \circ \cdots L_1 \circ \sigma, \quad L_t(w) = \sigma(\sigma^{-1}(M_t w) + K_t w + b_t),$$

where $\sigma \colon \mathbb{R} \to I$ and, for $t = 1, \ldots, T$, $K_t \in \mathbb{R}^{n_t \times n_{t-1}}$ and $b_t \in \mathbb{R}^{n_t}$, and for every $t = 1, \ldots, T-1$, $M_t \in \mathbb{R}^{n_t \times n_{t-1}}$, is right stochastic, $n_0 = J$ and $n_T = J'$. Since, we are assuming that Bregman neural networks contain constant functions (recall the sentence before Lemma 8), we have

- $b_t \mathbb{1}_D \in \mathsf{BN}_2(\sigma; \mathbb{R}^d, \mathbb{R}^{n_t})_{|\overline{D}} \subset \mathcal{C}(\overline{D}, \mathbb{R}^{n_t})$

- $\kappa_t = \dfrac{1}{|D|} K_t \mathbb{1}_{D \times D} \in \mathsf{BN}_2(\sigma; \mathbb{R}^d \times \mathbb{R}^d, \mathbb{R}^{n_t \times n_{t-1}})_{|\overline{D} \times \overline{D}} \subset \mathcal{C}(\overline{D} \times \overline{D}, \mathbb{R}^{n_t \times n_{t-1}})$ and

  $$\mathcal{K}_t \colon L^p(D, \mathbb{R}^{n_{t-1}}) \to L^q(D, \mathbb{R}^{n_t})$$
  $$v \mapsto (\mathcal{K}_t v)(x) = \int_D \kappa_t(x, y) v(y)\, dy = \int_D \frac{1}{|D|} K_t v(y)\, dy = K_t \bar{v},$$

  where $\bar{v}$ is the mean value of $v$. So that $\mathcal{K}_t v = (K_t \bar{v}) \mathbb{1}_D$ is a constant function.

- $\mu_t = \dfrac{1}{|D|} M_t \mathbb{1}_{D \times D} \in \mathsf{BN}_2(\sigma; \mathbb{R}^d \times \mathbb{R}^d, \mathbb{R}^{n_t \times n_{t-1}})_{|\overline{D} \times \overline{D}} \subset \mathcal{C}(\overline{D} \times \overline{D}, \mathbb{R}^{n_t \times n_{t-1}})$

  $$\mathcal{M}_t \colon L^p(D, \mathbb{R}^{n_{t-1}}) \to L^p(D, \mathbb{R}^{n_t})$$
  $$v \mapsto (\mathcal{M}_t v)(x) = \int_D \mu_t(x, y) v(y)\, dy = \int_D \frac{1}{|D|} M_t v(y)\, dy = M_t \bar{v}.$$

  Moreover, since $M_t$ is right stochastic, if the function $v$ has range (almost everywhere) in $I^{n_{t-1}}$, we have that $\bar{v} \in I^{n_{t-1}} \Rightarrow M_t \bar{v} \in I^{n_t}$, Hence

  $$\mathcal{M}_t(\operatorname{dom} \partial \Phi_{t-1}) \subset \operatorname{dom} \partial \Phi_t.$$

  Indeed, recall that $\Phi_t \colon L^p(D, \mathbb{R}^{n_t}) \to\, ]-\infty, +\infty]$ and

  $$\forall\, v \in L^p(D, \mathbb{R}^{n_t}) \colon\ \Phi_t(v) = \int_D \phi_t(v(x))\, dx, \quad \forall\, w \in \mathbb{R}^{n_t} \colon\ \phi_t(w) = \sum_{i=1}^{n_t} \psi(w_i)$$

  with $\psi \colon \mathbb{R} \to\, ]-\infty, +\infty]$ Legendre, $\operatorname{int}(\operatorname{dom}\psi) = I$, $\operatorname{dom}\psi^* = \mathbb{R}$, $\sigma = (\psi^*)'$, and $\sigma^{-1} = \psi'$, so that $\operatorname{dom} \partial \Phi_t = \{v \in L^p(D, \mathbb{R}^{n_t}) \mid \text{for a.e. } x \in D,\ v(x) \in I^{n_t}\}$ and for $v \in \operatorname{dom} \Phi_t$, $\partial \Phi_t(v) = \{\nabla \phi \circ v\}$.

It follows from the previous considerations that if $v \in \operatorname{dom} \partial \Phi_{t-1} \subset L^p(D, \mathbb{R}^{n_{t-1}})$, we have $\mathcal{K}_t(v) = (K_t \bar{v}) \mathbb{1}_D$ and $\mathcal{M}_t v = (M_t \bar{v}) \mathbb{1}_D$, and hence

$$\mathcal{L}_t(v) = \sigma(\sigma^{-1}(\mathcal{M}_t v) + \mathcal{K}_t v + b_t \mathbb{1}_D)(x) = \sigma(\sigma^{-1}(M_t \bar{v}) + K_t \bar{v} + b_t).$$

Note that here $\mathcal{V}_t = L^p(D, \mathbb{R}^{n_t})$. Thus, we have

$$\mathcal{L}_t(v) = (L_t \bar{v}) \mathbb{1}_D,$$

meaning that the operator layer $\mathcal{L}_t$ transforms any function in $L^p(D, \mathbb{R}^{n_t})$ into a constant function, where the constant is the mean value of the function, transformed via the standard (finite dimensional) Bregman layer $L_t$. In particular, if $w \in \mathbb{R}^J$, we have

$$\mathcal{L}_1(\sigma(w \mathbb{1}_D)) = \mathcal{L}_1(\sigma(w) \mathbb{1}_D) = L_1(\sigma(w)) \mathbb{1}_D$$
$$\mathcal{L}_2(\mathcal{L}_1(\sigma(w \mathbb{1}_D))) = \mathcal{L}_2(L_1(\sigma(w)) \mathbb{1}_D) = L_2(L_1(\sigma(w))) \mathbb{1}_D,$$

and so on. Therefore, if we set

$$\mathcal{N}^{\mathsf{BN}} = \mathcal{K}_T \circ \mathcal{L}_{T-1} \circ \cdots \circ \mathcal{L}_1 \circ \sigma,$$

the statement follows. $\qquad\square$

**Remark 12.** *Let $S\colon \mathbb{R}^{J'} \to \mathcal{U}(D, \mathbb{R}^k)$ be linear (and continuous) and set*

$$\forall\, i = 1, \ldots, J'\colon\; s_j = Se_j \in \mathcal{U},$$

*where $(e_j)_{1 \le j \le J'}$ is the canonical basis of $\mathbb{R}^{J'}$. Define the function $s\colon D \to \mathbb{R}^{k \times J'}$, with $s(x) = [s_1(x) \cdots s_J^T(x)]$, which has the $s_j$'s as columns. Then*

$$\forall\, w \in \mathbb{R}^{J'}\colon\; Sw = S\Big(\sum_{j=1}^{J'} w_j e_j\Big) = \sum_{j=1}^{J'} w_j s_j \;\Rightarrow\; (Sw)(x) = \sum_{j=1}^{J'} w_j s_j(x) = s(x)w.$$

*Thus, the action of $S$ can be represented by a matrix-valued function with columns in $\mathcal{U}$. Moreover, the linear operator $S$ can be lifted to a linear integral operator from $L^p(D, \mathbb{R}^{J'})$ to $\mathcal{U}$. Indeed if we define the kernel*

$$\kappa_s\colon D \times D \to \mathbb{R}^{k \times J'}, \quad \kappa_s(x, y) = \frac{1}{|D|} s(x),$$

*for every $v \in L^p(D, \mathbb{R}^{J'})$, we have*

$$(\mathcal{S}v)(x) = \int_D \kappa_s(x, y) v(y)\, dy = \int_D \frac{1}{|D|} s(x) v(y)\, dy = s(x)\bar{v},$$

*where $\bar{v}$ is the mean value of $v$. In the end $\mathcal{S}\colon L^p(D, \mathbb{R}^{J'}) \to \mathcal{U}$ and*

$$\forall\, v \in L^p(D, \mathbb{R}^{J'})\colon\; \mathcal{S}v = S\bar{v},$$

*and hence, for every $w \in \mathbb{R}^{J'}$, $\mathcal{S}(w\mathbb{1}_D) = Sw$, meaning that $\mathcal{S}$ is actually an extension of $S$ to the Lebesgue space $L^p(D, \mathbb{R}^{J'})$.*

**Lemma 11.** *Let $S\colon \mathbb{R}^{J'} \to \mathcal{U}(D, \mathbb{R}^k)$ be linear (and continuous). Let $K \subset \mathbb{R}^{J'}$ be a compact set and $\varepsilon > 0$. Then there exists a function $h \in \mathsf{BN}_2(\sigma; \mathbb{R}^d, \mathbb{R}^{k \times J'})_{|D}$ so that for the corresponding linear operator $S^{\mathsf{BN}}\colon \mathbb{R}^{J'} \to \mathcal{U}$ defined as*

$$\forall\, w \in \mathbb{R}^{J'}\colon\; (S^{\mathsf{BN}}w)(x) = \sum_{i=1}^{J'} w_j h_j(x) = h(x)w,$$

*according to Remark 12, we have*

$$\sup_{w \in K} \|Sw - S^{\mathsf{BN}}w\|_{\mathcal{U}} < \varepsilon.$$

Finally we are ready for the proof of Theorem 3.

*Proof of Theorem 3.* It follows from Fact 3 that there exist $J, J' \in \mathbb{N}$ and

$$R\colon \mathcal{A} \to \mathbb{R}^J, \quad f\colon \mathbb{R}^J \to \mathbb{R}^{J'}, \quad S\colon \mathbb{R}^{J'} \to \mathcal{U},$$

with $R$ and $S$ linear continuous and $f$ continuous, such that

$$\sup_{v \in K} \|\mathcal{G}(v) - (S \circ f \circ R)(v)\| < \varepsilon.$$

Now, taking advantage of the previous lemmas we want to replace the operators $R$ and $S$ with analogue operators depending on shallow Bregman neural networks, and the function $f$ with a Bregman neural network. It follows from Lemma 9 that for every $n \in \mathbb{N}$ there exist

$$R_n^{\mathsf{BN}}\colon \mathcal{A} \to \mathbb{R}^J \text{ linear continuous operator such that } \sup_{v \in K} |Rv - R_n^{\mathsf{BN}}v| < \frac{1}{n+1},$$

where $R_n^{\mathsf{BN}}$ depends on a Bregman shallow network $h_n$ as specified in Lemma 9. Clearly this implies that $\lim_{n \to +\infty} R_n^{\mathsf{BN}}v = Rv$ uniformly on $K$, so that the set

$$K_1 := R(K) \cup \bigcup_{n \in \mathbb{N}} R_n^{\mathsf{BN}}(K) \subset \mathbb{R}^J$$

is compact (see Kovachki et al. (2023, Lemma 21)). Since $f$ is continuous, it is uniformly continuous on $K_1$, hence given $\varepsilon > 0$ there exists $\delta > 0$ such that

$$\forall\, w, w' \in K_1 \colon\ |w - w'| < \delta\ \Rightarrow\ |f(w) - f(w')| < \frac{\varepsilon}{3\,\|S\|}.$$

Moreover, there exists $f^{\mathsf{BN}} \in \mathsf{BN}_2(\sigma; \mathbb{R}^J, \mathbb{R}^{J'})$ such that

$$\sup_{w \in K_1} |f(w) - f^{\mathsf{BN}}(w)| < \frac{\varepsilon}{3\,\|S\|}.$$

Let's take $n \in \mathbb{N}$ such that $1/(n+1) < \delta$. Then,

$$\forall\, v \in K \colon\ \ Rv, R_n^{\mathsf{BN}} v \in K_1 \text{ and } |Rv - R_n^{\mathsf{BN}} v| < \frac{1}{n+1} < \delta\ \Rightarrow\ |f(Rv) - f(R_n^{\mathsf{BN}} v)| < \frac{\varepsilon}{3\,\|S\|}.$$

Finally, since $f^{\mathsf{BN}}(K_1)$ is compact, by Lemma 11, there exist $S^{\mathsf{BN}} \colon \mathbb{R}^{J'} \to \mathcal{U}$ such that

$$\sup_{w \in f^{\mathsf{BN}}(K_1)} \|Sw - S^{\mathsf{BN}} w\|_{\mathcal{U}} < \frac{\varepsilon}{3}.$$

Therefore, for every $v \in K$ we have

$$\begin{aligned}
\left\| S(f(Rv)) - S^{\mathsf{BN}}(f^{\mathsf{BN}}(R_n^{\mathsf{BN}} v)) \right\|_{\mathcal{U}} &\le \left\| S(f(Rv)) - S(f(R_n^{\mathsf{BN}} v)) \right\|_{\mathcal{U}} + \left\| S(f(R_n^{\mathsf{BN}} v)) - S(f^{\mathsf{BN}}(R_n^{\mathsf{BN}} v)) \right\|_{\mathcal{U}} \\
&\quad + \left\| S(f^{\mathsf{BN}}(R_n^{\mathsf{BN}} v)) - S^{\mathsf{BN}}(f^{\mathsf{BN}}(R_n^{\mathsf{BN}} v)) \right\|_{\mathcal{U}} \\
&\le \|S\|\, |f(Rv) - f^{\mathsf{BN}}(R_n^{\mathsf{BN}} v)| + \|S\|\, |f(R_n^{\mathsf{BN}} v) - f^{\mathsf{BN}}(R_n^{\mathsf{BN}} v)| \\
&\quad + \left\| S(f^{\mathsf{BN}}(R_n^{\mathsf{BN}} v)) - S^{\mathsf{BN}}(f^{\mathsf{BN}}(R_n^{\mathsf{BN}} v)) \right\|_{\mathcal{U}} \\
&< \frac{\varepsilon}{3} + \frac{\varepsilon}{3} + \frac{\varepsilon}{3} = \varepsilon.
\end{aligned}$$

In the end, for every $v \in K$,

$$\left\| \mathcal{G}(v) - S^{\mathsf{BN}}(f^{\mathsf{BN}}(R_n^{\mathsf{BN}} v)) \right\|_{\mathcal{U}} \le \|\mathcal{G}(v) - S(f(Rv))\|_{\mathcal{U}} + \left\| S(f(Rv)) - S^{\mathsf{BN}}(f^{\mathsf{BN}}(R_n^{\mathsf{BN}} v)) \right\|_{\mathcal{U}} < 2\varepsilon.$$

Now in order to conclude the proof, it is sufficient to lift the operators $R^{\mathsf{BN}}$ and $S^{\mathsf{BN}}$ to Lebesgue spaces, as described in Remark 11 and Remark 12, and the function $f^{\mathsf{BN}}$ to Bregman neural operator as described in Lemma 10 and recognize that

$$\mathcal{S}^{\mathsf{BN}} \circ \mathcal{N}^{\mathsf{BN}} \circ \mathcal{R}_n^{\mathsf{BN}} = S^{\mathsf{BN}} \circ f^{\mathsf{BN}} \circ R_n^{\mathsf{BN}}.$$

Indeed, for every $v \in \mathcal{A}$, we have

$$\mathcal{S}^{\mathsf{BN}}(\mathcal{N}^{\mathsf{BN}}(\mathcal{R}_n^{\mathsf{BN}} v)) = \mathcal{S}^{\mathsf{BN}}(\mathcal{N}^{\mathsf{BN}}((R_n^{\mathsf{BN}} v)\mathbb{1}_D)) = \mathcal{S}^{\mathsf{BN}}(f^{\mathsf{BN}}((R_n^{\mathsf{BN}} v))\mathbb{1}_D) = S^{\mathsf{BN}}(f^{\mathsf{BN}}((R_n^{\mathsf{BN}} v))).$$

The statement follows. $\qquad\qquad\square$

## C  EXPERIMENTAL SETTINGS

We adopt the same experimental setting as in the PDEBench repository (Takamoto et al., 2022). For the sake of information, we recall the considered problems and PDEs and the specific settings we consider when appropriate. The learning procedure used is presented at the end of this section.

### C.1  1D ADVECTION EQUATION

The advection equation is a linear Partial Differential Equation (PDE) modeling the transport of a fluid quantity $u$, namely its velocity field, defined by the following equation:

$$\partial_t u(x,t) + \beta \partial_x u(x,t) = 0, \quad x \in (0,1), t \in (0,2], \tag{20}$$

$$u(x,0) = u_0(x), \ x \in (0,1), \tag{21}$$

with $\beta$ a constant advection speed. Note that this system admits an exact solution: $u(t,x) = u_0(x - \beta t)$.

For this dataset, we follow the setting given in Takamoto et al. (2022), Section D.1 by taking $\beta = 0.4$. We learn the mapping between the value of the field at $t = 0$ ($u(x,0)$) and the value at time $t = 2$ ($u(x,2)$), *i.e.* we learn the mapping between the first and the last temporal value of each sample.

### C.2  1D BURGERS EQUATION

The Burgers' equation is a PDE describing the nonlinear advection and diffusion of a velocity field, defined as follows:

$$\partial_t u(x,t) + \partial_x(u^2(x,t)/2) = \nu/\pi \partial_{xx} u(x,t), \quad x \in (0,1), t \in (0,2], \tag{22}$$

$$u(x,0) = u_0(x), \ x \in (0,1), \tag{23}$$

where $\nu$ is the diffusion coefficient, which is assumed to be constant in this dataset.

We follow again the setup presented in Takamoto et al. (2022), section D.2, with $\nu = 0.001$. As in the previous dataset, we learn the mapping from the field at $t = 0$ as input to the field at $t = 2$ as target.

### C.3  1D COMPRESSIBLE NAVIER-STOKES EQUATIONS (1D NS)

The compressible Navier-Stokes equations describe the motion of viscous fluids that can change in density due to compression or expansion. This can be described through the following partial differential equations:

$$\partial_t \sigma + \partial_x \cdot (\sigma \mathbf{u}) = 0, \tag{24}$$

$$\sigma(\partial_t \mathbf{u} + \mathbf{u} \cdot \partial_x \mathbf{u}) = -\partial_x p + \eta \triangle \mathbf{u} + (\zeta + \eta/3)\partial_{xx}\mathbf{u}), \tag{25}$$

$$\partial_t(\epsilon + \sigma v^2/2) + \partial_x \cdot [(p + \epsilon + \sigma v^2/2)\mathbf{u} - \mathbf{u} \cdot \sigma'] = \mathbf{0}, \tag{26}$$

where $\sigma$ is the mass density, $\mathbf{u} = \mathbf{u}(\mathbf{x}, \mathbf{t})$ is the fluid velocity, $p$ is the gas pressure, $\epsilon$ is an internal energy described by the equation of state, $\sigma'$ is the viscous stress tensor, and $\eta$ and $\zeta$ are shear and bulk viscosity, respectively.

In our experiments, we consider the setup introduced in Takamoto et al. (2022), Section D.5, fixing $\eta = 10^{-8}$, $\zeta = 10^{-8}$ and out-going boundary conditions. We learn the mapping of the velocity $\mathbf{v}$ from time $t = 10$ as input to time $t = 15$ as target. For this dataset, we added a symmetrical padding preprocessing to replicate periodic boundary conditions (as prescribed in the original FNO code (Li et al., 2021a)).

### C.4  2D INCOMPRESSIBLE NAVIER-STOKES EQUATIONS (2D NS)

We also consider a dataset from the 2D Navier-Stokes equation for a viscous, incompressible fluid in vorticity form on the unit torus (Li et al., 2021a) defined as follows:

$$\partial_t w(x,t) + u(x,t) \cdot \nabla w(x,t) = \nu \Delta w(x,t) + f(x), \qquad x \in (0,1)^2, t \in (0, T_{final}]$$

$$\nabla \cdot u(x,t) = 0, \qquad x \in (0,1)^2, t \in (0, T_{final}] \tag{27}$$

$$w(x,0) = w_0(x), \qquad x \in (0,1)^2$$

with $u$ is the 2D velocity field, $w = \nabla \times u$ is the vorticity, $w_0 : (0,1)^2; \to \mathbb{R}$ is the initial vorticity function, $\nu \in \mathbb{R}_+$ is the viscosity coefficient, and $f : (0,1)^2 \to \mathbb{R}$ is the forcing function.

We follow the setup introduced in Li et al. (2021a), Section A.3.3, with $\nu = 10^{-3}$ and $\nu = 10^{-4}$. We learn the mapping of the velocity field $\mathbf{v}$ from sample time $t = 10$ to $t = 50$ for $\nu = 10^{-3}$ and from $t = 10$ to $t = 20$ for $\nu = 10^{-4}$.

## C.5 DARCY FLOW

We consider a dataset based on the steady state of the 2D Darcy Flow equation on the unit square, representing the flow through porous media and defined as follows:

$$
\begin{aligned}
-\nabla(a(x)\nabla u(x)) &= f(x), & x &\in (0,1)^2, \\
u(x) &= 0, & x &\in \partial(0,1)^2.
\end{aligned}
\tag{28}
$$

We follow the setup described in Takamoto et al. (2022), Section D.4, with $f(x)$ fixed to the constant $\beta = 0.1$.

## C.6 LEARNING PROCEDURE

Models are trained using the Adam optimizer with a constant learning rate, a batch size of 128 for 1D problems (resp. 32 for 2D problems), a maximum of 2000 epochs and an early stopping strategy with patience of 100 epochs and $\delta = 10^{-3}$. The learning rate is validated on a grid of multiple values equally spaced in logarithmic scale. If not mentioned otherwise, we use 8000 (resp. 1000) training samples for 1D (resp. 2D) problems, and 1000 samples each for validation and testing. All results are averaged over four random splittings.

Experiments have been made on an internal clusters of GPUs with memory from 10Go to 45Go. All the experiments can be achieved with GPUs with a memory of 10Go, except for models with 32 or 64 layers which require at least a memory of 24Go.

# D ADDITIONAL RESULTS

## D.1 COMPARISON OF PREDICTIONS

In this section, we visually inspect to what extent the prediction made by FNO and BFNO is close to the ground truth. We provide two examples on two different datasets: 1D Burgers (ground truth in dashed black) and 2D Darcy (ground truth "*Output u*").
As discussed in the subsequent analysis (see Appendix D.3), BFNO better learn the higher frequencies, as shown by the sharper edges closer to the ground truth.

FIX

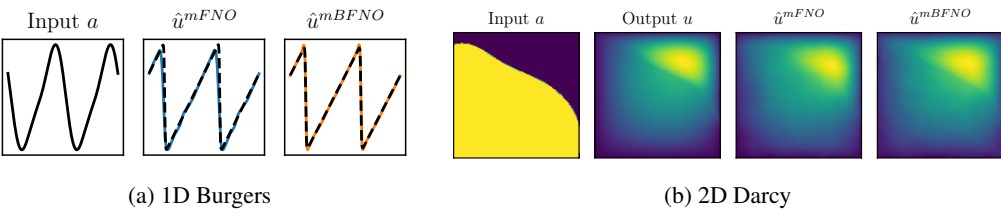

(a) 1D Burgers  (b) 2D Darcy

Figure 6: Visual Comparison of Prediction.

## D.2 EXTENSION TO WNOS AND ANALYSIS UP TO 64 LAYERS

In the following section, we provide additional results comparing models with 32 and 64 layers.

At this point, exploding gradients can be a common issue. To avoid it, we applied gradient clipping for all 32 and 64-layer models. As seen in Table 3, we observe a decrease in performance of standard FNOs, confirming our observations with 8 and 16 layers. However, our models show some improvement when increasing further the number of layers.

We also extended our experiments to Wavelet Neural Operators (WNO). In Table 4 is reported the comparison between standard WNO and the Bregman version BWNO. We can observe similar results as Fourier models, where our models outperform the standard models and are able to gain performance when increasing the number of layers. Furthermore, even with gradient clipping, 32 and 64-layer standard models could not converge during training, leading to 100% relative error rate. Further analysis shows that this divergence can be linked with the high error rates on low frequencies and boundary conditions.

Table 3: Relative error of FNO and BFNO models on benchmark PDEs.

|  | 1D Burgers | | 1D NS | | 2D Darcy | |
|  | **FNO** | **BFNO** | **FNO** | **BFNO** | **FNO** | **BFNO** |
| --- | --- | --- | --- | --- | --- | --- |
| **4 layers** | $5.5 \pm 0.1\%$ | $\mathbf{5.4 \pm 0.1\%}$ | $58.2 \pm 0.6\%$ | $\mathbf{57.0 \pm 0.6\%}$ | $34.6 \pm 0.0\%$ | $\mathbf{33.4 \pm 0.2\%}$ |
| **8 layers** | $5.4 \pm 0.1\%$ | $\mathbf{4.1 \pm 0.2\%}$ | $58.2 \pm 0.6\%$ | $\mathbf{56.8 \pm 0.8\%}$ | $32.8 \pm 0.2\%$ | $\mathbf{31.5 \pm 0.4\%}$ |
| **16 layers** | $6.5 \pm 0.1\%$ | $\mathbf{3.5 \pm 0.2\%}$ | $59.7 \pm 0.6\%$ | $\mathbf{56.5 \pm 0.6\%}$ | $32.9 \pm 0.2\%$ | $\mathbf{30.0 \pm 0.5\%}$ |
| **32 layers** | $10.2 \pm 0.8\%$ | $\mathbf{3.5 \pm 0.2\%}$ | $71.5 \pm 0.6\%$ | $\mathbf{55.7 \pm 0.5\%}$ | $35.6 \pm 0.4\%$ | $\mathbf{29.6 \pm 0.4\%}$ |
| **64 layers** | $11.1 \pm 0.7\%$ | $\mathbf{3.4 \pm 0.2\%}$ | $74.7 \pm 1.2\%$ | $\mathbf{56.0 \pm 0.6\%}$ | $36.9 \pm 1.0\%$ | $\mathbf{29.0 \pm 0.2\%}$ |

Table 4: Relative error of WNO and BWNO models on benchmark PDEs.

|  | 1D Advection | | 1D Burgers | | 1D NS | |
|  | **WNO** | **BWNO** | **WNO** | **BWNO** | **WNO** | **BWNO** |
| --- | --- | --- | --- | --- | --- | --- |
| **4 layers** | $3.0 \pm 0.0\%$ | $\mathbf{2.8 \pm 0.2\%}$ | $21.5 \pm 0.5\%$ | $\mathbf{21.3 \pm 0.4\%}$ | $59.2 \pm 0.6\%$ | $\mathbf{58.3 \pm 0.6\%}$ |
| **8 layers** | $2.5 \pm 0.1\%$ | $\mathbf{2.1 \pm 0.1\%}$ | $19.1 \pm 0.6\%$ | $\mathbf{17.9 \pm 0.6\%}$ | $59.0 \pm 0.6\%$ | $\mathbf{58.0 \pm 0.6\%}$ |
| **16 layers** | $3.9 \pm 0.8\%$ | $\mathbf{2.0 \pm 0.2\%}$ | $19.7 \pm 0.5\%$ | $\mathbf{16.4 \pm 0.3\%}$ | $61.1 \pm 0.6\%$ | $\mathbf{57.6 \pm 0.7\%}$ |
| **32 layers** | $100 \pm 0\%$ | $\mathbf{1.9 \pm 0.1\%}$ | $100 \pm 0\%$ | $\mathbf{16.4 \pm 0.5\%}$ | $100 \pm 0\%$ | $\mathbf{57.2 \pm 0.6\%}$ |
| **64 layers** | $100 \pm 0\%$ | $\mathbf{1.8 \pm 0.2\%}$ | $100 \pm 0\%$ | $\mathbf{16.1 \pm 0.4\%}$ | $100 \pm 0\%$ | $\mathbf{57.5 \pm 0.6\%}$ |

### D.3 Detailed Analysis of the Prediction Performance

In the same spirit of Takamoto et al. (2022), we include several metrics providing a deeper understanding of the models' behavior, including relative mean squared error on the boundary (rMSE) as well as in the low, mid, and high frequency bands (fRMSE low, fRMSE mid, fRMSE high). Results are provided in Table 5.

FIX

Table 5: Additional comparison of the performance in terms of relative $\ell^2$ error (rL2), relative mean squared error on the boundary (rMSE) as well as in the low, mid and high frequency bands (fRMSE low, fRMSE mid, fRMSE high). Note that here 2D NS corresponds to $\nu = 10^{-3}$.

| PDE | Metric | $T = 4$ | | $T = 8$ | | $T = 16$ | |
|---|---|---|---|---|---|---|---|
| | | BFNO | FNO | BFNO | FNO | BFNO | FNO |
| 1D advection | rL2 | $1.03 \cdot 10^{-2}$ | $6.82 \cdot 10^{-3}$ | $6.43 \cdot 10^{-3}$ | $1.36 \cdot 10^{-2}$ | $6.43 \cdot 10^{-3}$ | $1.81 \cdot 10^{-2}$ |
| | bRMSE | $1.14 \cdot 10^{-1}$ | $1.62 \cdot 10^{0}$ | $1.16 \cdot 10^{-1}$ | $3.87 \cdot 10^{0}$ | $1.12 \cdot 10^{-1}$ | $1.58 \cdot 10^{1}$ |
| | fRMSE low | $7.10 \cdot 10^{-6}$ | $7.37 \cdot 10^{-5}$ | $7.59 \cdot 10^{-6}$ | $2.33 \cdot 10^{-4}$ | $7.62 \cdot 10^{-6}$ | $1.36 \cdot 10^{-3}$ |
| | fRMSE mid | $5.41 \cdot 10^{-6}$ | $1.77 \cdot 10^{-5}$ | $5.07 \cdot 10^{-6}$ | $3.78 \cdot 10^{-5}$ | $4.65 \cdot 10^{-6}$ | $1.08 \cdot 10^{-4}$ |
| | fRMSE high | $4.20 \cdot 10^{-7}$ | $2.03 \cdot 10^{-6}$ | $3.60 \cdot 10^{-7}$ | $3.20 \cdot 10^{-6}$ | $3.60 \cdot 10^{-7}$ | $5.18 \cdot 10^{-6}$ |
| 1D Burgers | rL2 | $5.37 \cdot 10^{-2}$ | $5.48 \cdot 10^{-2}$ | $4.14 \cdot 10^{-2}$ | $5.42 \cdot 10^{-2}$ | $3.51 \cdot 10^{-2}$ | $6.45 \cdot 10^{-2}$ |
| | bRMSE | $4.31 \cdot 10^{-1}$ | $4.38 \cdot 10^{-1}$ | $2.97 \cdot 10^{-1}$ | $3.80 \cdot 10^{-1}$ | $2.39 \cdot 10^{-1}$ | $4.79 \cdot 10^{-1}$ |
| | fRMSE low | $5.67 \cdot 10^{-5}$ | $5.70 \cdot 10^{-5}$ | $3.63 \cdot 10^{-5}$ | $5.08 \cdot 10^{-5}$ | $3.08 \cdot 10^{-5}$ | $5.31 \cdot 10^{-5}$ |
| | fRMSE mid | $3.49 \cdot 10^{-5}$ | $3.44 \cdot 10^{-5}$ | $2.70 \cdot 10^{-5}$ | $3.58 \cdot 10^{-5}$ | $2.38 \cdot 10^{-5}$ | $3.79 \cdot 10^{-5}$ |
| | fRMSE high | $1.17 \cdot 10^{-6}$ | $1.20 \cdot 10^{-6}$ | $1.07 \cdot 10^{-6}$ | $1.24 \cdot 10^{-6}$ | $9.80 \cdot 10^{-7}$ | $1.23 \cdot 10^{-6}$ |
| 2D NS | rL2 | $4.27 \cdot 10^{-2}$ | $4.61 \cdot 10^{-2}$ | $4.01 \cdot 10^{-2}$ | $4.14 \cdot 10^{-2}$ | $3.98 \cdot 10^{-2}$ | $3.90 \cdot 10^{-2}$ |
| | bRMSE | $3.87 \cdot 10^{-2}$ | $4.16 \cdot 10^{-2}$ | $3.63 \cdot 10^{-2}$ | $3.76 \cdot 10^{-2}$ | $3.61 \cdot 10^{-2}$ | $3.54 \cdot 10^{-2}$ |
| | fRMSE low | $4.05 \cdot 10^{-4}$ | $4.33 \cdot 10^{-4}$ | $3.72 \cdot 10^{-4}$ | $3.82 \cdot 10^{-4}$ | $3.80 \cdot 10^{-4}$ | $3.63 \cdot 10^{-4}$ |
| | fRMSE mid | $9.59 \cdot 10^{-5}$ | $9.03 \cdot 10^{-5}$ | $6.53 \cdot 10^{-5}$ | $7.73 \cdot 10^{-5}$ | $6.22 \cdot 10^{-5}$ | $6.15 \cdot 10^{-5}$ |
| | fRMSE high | $9.85 \cdot 10^{-6}$ | $6.95 \cdot 10^{-6}$ | $5.95 \cdot 10^{-6}$ | $5.98 \cdot 10^{-6}$ | $5.40 \cdot 10^{-6}$ | $9.18 \cdot 10^{-6}$ |

### D.4 Comparison with Other FNO baselines

In this section, we provide comparisons with other FNO improvements on the 2D Navier-Stokes dataset with $\nu = 10^{-4}$. In particular, we consider F-FNO (Tran et al., 2023) which is a particularly relevant baseline for comparison, as it i) incorporates skip-like connections that share similarities with our additional $\sigma^{-1}$ term and ii) also seeks to enable the development of deeper FNO architectures.

We consider the best-performing F-FNO model (as identified by its authors), trained using our optimization strategy and adapted to our specific learning task. It is important to note that F-FNO was originally designed for predicting mappings between multiple consecutive time steps (e.g., from $t$ to $t + 1$) and it offers the option to rely on techniques such as the Markov assumption and teacher forcing. Since our task involves predicting the final state directly from the initial conditions, those techniques are not appropriate, and thus we did not include them in the implementation. Additionally, beyond the optimization strategy proposed by the F-FNO authors (AdamW with cosine annealing, noise injection and input normalization), we also employed the optimization strategy detailed in Appendix C.6. In the following, we name them F-FNO$_{\text{base}}$ and F-FNO, respectively.
Results reported in Table 6 show that the original training procedure of the F-FNO does not transfer well to our learning task, as evidenced by the poor performance of F-FNO$_{\text{base}}$ across all layers, compared to the other models. On the contrary, with our training strategy and adapted hyperparameters, F-FNO shows strong performance, especially with few layers.

Moreover, to isolate the impact of residual connections from the broader structural modifications introduced by F-FNO, we have also implemented and compared a ResNet-inspired variant of FNO, referred to as ResFNO. We did this to better understand the role of the residual connection.
As shown in Table 6, ResFNO consistently achieves lower training error than FNO, which aligns with the behavior typically observed in ResNet-like architectures. However, it falls short in terms of generalization to unseen data, as reflected in the test error.

In conclusion, BFNO demonstrates superior scalability than its competitors (see for 16 layers), evidencing that the additional $\sigma^{-1}$ term, inherent to our Bregman model, better facilitates the training and generalization of deeper architectures.

NEW

Table 6: Comparison of train and test relative $\ell^2$ error across different architectures and number of layers.

| Metric | Model | 4 layers | 8 layers | 16 layers |
|---|---|---|---|---|
| | FNO | $13.5 \pm 0.1$ | $13.0 \pm 0.2$ | $12.6 \pm 0.1$ |
| | ResFNO | $13.5 \pm 0.1$ | $12.8 \pm 0.1$ | $13.0 \pm 0.1$ |
| | F-FNO$_{base}$ | $15.4 \pm 0.3$ | $14.9 \pm 0.4$ | $15.1 \pm 0.2$ |
| Test $\ell^2$ (%) | F-FNO | $\mathbf{13.0 \pm 0.1}$ | $\mathbf{12.4 \pm 0.1}$ | $12.7 \pm 0.2$ |
| | BFNO | $13.7 \pm 0.1$ | $12.6 \pm 0.1$ | $\mathbf{12.2 \pm 0.1}$ |
| | FNO | $4.6 \pm 0.2$ | $3.6 \pm 0.1$ | $4.0 \pm 0.1$ |
| | ResFNO | $\mathbf{3.7 \pm 0.2}$ | $\mathbf{2.9 \pm 0.1}$ | $\mathbf{2.8 \pm 0.1}$ |
| Train $\ell^2$ (%) | F-FNO$_{base}$ | $10.4 \pm 0.3$ | $8.7 \pm 0.3$ | $7.3 \pm 0.1$ |
| | F-FNO | $4.1 \pm 0.0$ | $3.3 \pm 0.5$ | $4.0 \pm 0.4$ |
| | BFNO | $4.4 \pm 0.3$ | $3.7 \pm 0.3$ | $3.4 \pm 0.2$ |

## D.5 IMPACT OF THE ACTIVATION FUNCTION

A limitation of our framework is the fact that it requires Bregman variants (such as BFNO) to have a *strictly* monotonic activation function, which excludes a few functions such as ReLU. This justifies why in our experiments we used Softplus as a surrogate of ReLU. On the contrary, for classical neural operators within our framework, the activation function only needs to be monotonic, not strictly monotonic. Therefore, ReLU is still valid and can be used.

As a thought experiment, we also implemented BFNO with ReLU and evaluated it on the 2D Navier-Stokes dataset ($\nu = 10^{-4}$). Table 7 shows that BFNO with ReLU achieves comparable or better performance than Softplus for the same number of layers. However, the best results are the same (i.e., 12.2% for 16 layers).

NEW

Table 7: Comparison BFNO with Softplus or ReLU.

| Architecture | 4 layers | 8 layers | 16 layers |
|---|---|---|---|
| FNO (ReLU) | $13.5 \pm 0.1$ | $13.0 \pm 0.1$ | $12.6 \pm 0.1$ |
| BFNO (Softplus) | $13.7 \pm 0.1$ | $12.6 \pm 0.1$ | $\mathbf{12.2 \pm 0.1}$ |
| BFNO (ReLU) | $\mathbf{13.4 \pm 0.2}$ | $\mathbf{12.2 \pm 0.2}$ | $\mathbf{12.2 \pm 0.1}$ |

## D.6 IMPACT OF BATCH NORMALIZATION

For all the experiments presented in the previous sections, we relied on the latest available version of the FNO implementation, which does not include *Batch Normalization* (BN), while it was used in the original FNO paper Li et al. (2021a). We note that the original FNO code was removed from the GitHub repository by its author (i.e., the 'master' branch was deleted). While we retrieved an earlier version of the code, we observed that BN was implemented in the initial commit but was subsequently removed in a later commit titled "*remove unnecessary batchnorm*", suggesting that adding BN layers does not lead to better prediction performance.

To complement our results, we have conducted an experiment with BN for both FNO and BFNO architectures on the 2D Navier-Stokes dataset ($\nu = 10^{-4}$). Results, reported in Table 8, show marginal improvements for 8-layer models (FNO: 13.0% $\rightarrow$ 12.8%, BFNO: 12.6% $\rightarrow$ 12.4%) but no consistent benefits for other configurations. This aligns with the conclusion of the recent FNO implementations that removed BN.

NEW

Table 8: Impact of BatchNormalization (BN) with FNO and BFNO

| Architecture | 4 layers | 8 layers | 16 layers |
|---|---|---|---|
| FNO | **13.5 ± 0.1** | 13.0 ± 0.1 | 12.6 ± 0.1 |
| FNO + BN | **13.5 ± 0.1** | 12.8 ± 0.2 | 12.6 ± 0.1 |
| BFNO | 13.7 ± 0.1 | 12.6 ± 0.1 | **12.2 ± 0.1** |
| BFNO + BN | **13.5 ± 0.1** | **12.4 ± 0.0** | 12.3 ± 0.1 |

## D.7 WEIGHT DISTRIBUTION

We now present an analysis of the probability density function (PDF) of the weights for FNO and BFNO, trained on the Burgers dataset. The results, shown in Figure 7, reveal distinct behaviors between the two models. While the FNO PDF follows a Gaussian distribution, the BFNO PDF is more sharply peaked around 0, resembling a Laplacian distribution.

FIX

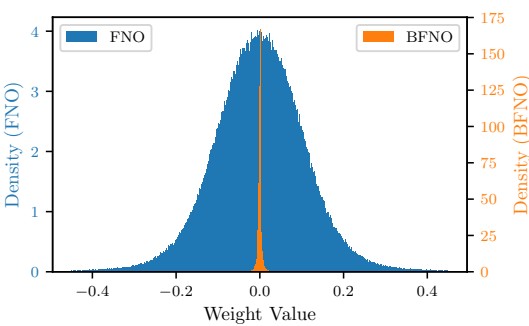

Figure 7: Illustration of the weights probability density function for FNO and BFNO models.

