# OpenReview forum: "A Bregman Proximal Viewpoint on Neural Operators"
_ICLR.cc/2025/Conference — ICLR 2025 Conference Withdrawn Submission_

### Official Review · Reviewer_n7Dc · 2024-10-27

**Soundness:** 2
**Presentation:** 1
**Contribution:** 1
**Rating:** 3
**Confidence:** 4

**Summary:**

This is a primarily theoretical paper which proposes to address a lack of formalism for characterizing architecture in previous methods.  The authors propose to replace the basic layer of a Fourier Neural Operator, equation (3), with a more general layer, equation first written down in equation (4).  This architecture was actually originally proposed in Frecon et al. (2022) "Bregman neural networks".  A large part of the papers introduces technical theory from convex analysis.  University approximation results are presented for the architecture.  Numerical experiments are presented comparing FNO with BFNO.  Seven equations are solved. It is demonstrated Fig 2a, that on the Burgers equation, the accuracy of FNO degrades as the number of layers are increased while BFNO is more accurate.

**Strengths:**

The authors bring the formalism of Bregman operators, which is normally used in convex analysis and optimization to bear on the analysis of neural network architectures.

**Weaknesses:**

The paper introduced a heavy mathematical formalism without justification.  The ideas of the paper are not clearly presented and the theoretical contribution is not substantial.

Overly technical:

- General conference audience will not understand the technical papers
- Neural operator specialists: will not understand the paper.
- Only experts in convex analysis will be able to follow much of the paper.
- Section 2 begins with a dense paragraph of convex analysis, unreadable to anyone who doesn't already know the area.  It's also unconnected to later sections, so not immediatly clear what is needed from this paragraph.
- Section 2.3 is an overview of bregman operators, bregman distance, textbook material, again not clear how much is needed.  "For additional details, the reader canrefer to Bauschke & Combettes (2017)"

The architecture is not clearly defined in the paper.
- The architecture was originally proposed in  Frecon et al. (2022) "Bregman neural networks" which was not an influential paper.  In that paper, it was made clear that the architecture involves bilevel optimization.  This is not explained or made clear in the current paper.
- In section 2.2. equation (4), which is a modification of (3), names the architecture, but it is not defined.
Later, it is defined as a special case of (6), which takes half a page to write down. "In (6), the proximity operator plays the role of an activation function operator, which in general will have the form of a nonlinear Nemytskii operator. ... Moreover, differently from usual architectures (Kovachki et al., 2021), ..."
- Nemytskii operator is never defined. However, "This relationship is crucial for further establishing connections with neural operator layers."

Experiments:
The numerical experiments are quite limited and show minimal improvement, or improvement in very particular situations, compared to FNO.
"To this end, we conducted an experiment using the Burgers’ dataset with viscosity ν = 10−3, with results presented in Figure 2a. First, we observe that BFNO systematically yields lower prediction error, irrespectively of T. Second, the performanceof FNO degrades starting from T = 16, while BFNO demonstrates better performance as T increases until it reaches a plateau at T = 64. "

- Was this also the case for the other 6 equations solved, or just for this one?  Why did you just present the one equation

**Questions:**

Questions:

1. Explain the architecture in more direct way.  Is there a bilevel optimization? This was much easier to understand in Frecon et al. (2022) "Bregman neural networks"
2. "Previous methods often lack of a general formalism for characterizing their architecture."  But there is a well known paper that does:  "Neural Operator: Learning Maps Between Function Spaces.  How does the contribution of this paper related to that one?
3. "In this work, we propose a novel expressive framework for neural operators by conceptualizing the action of operator layers as the minimizers of Bregman regularized optimization problems over Banach function spaces."  This sentence does not make sense, please clarify.

4. "We prove universal approximation results". This is true of almost any reasonable neural network, including MLP.  How does this argument show anything special about this particular architecture?

- "the proposed framework allows applying the extensive body of literature on proximal numerical optimization, of which Bregman proximity operators belong to, in order to study neural operators."
  - convince me why this is useful.
- "This opens the way to extend the analysis done on neural networks to (Bregman) neural operators in the same spirit of Combettes & Pesquet (2020a;b)."
  - Explain what would be achieved by this.
5. In Fig 2a, did the same results hold the other 6 equations solved, or just for this one?  Why did you just present the results for Burgers equation?

---

> ### Author Response · Authors · 2024-11-26
>
> 1. **Explain the architecture in a more direct way. Is there a bilevel optimization? This was much easier to understand in Frecon et al. (2022) "Bregman neural networks"**
>
>    We would like to start with a few preliminary facts.
>    First, our work is a generalization of the paper "Bregman neural networks" to infinite dimensional function spaces which additionally provides universal approximation results, that were not present in the original work by Frecon et al. (2022).
>    Second, similar to Frecon et al. (2022), once each layer is viewed as an optimization problem, the overall training of the architecture can be seen as a bilevel optimization problem, where at the lower level appear the optimization problems defining the action of each layer. Therefore, such bilevel formulation of the training phase is also shared by our proposal.
>    Third, extending the work by Frecon et al. (2022) to handle neural operators between (infinite dimensional) functional spaces is not an easy task and some level of mathematical formalism is unavoidable, since infinite dimensional problems, when properly addressed, inherently require a more complex and delicate analysis. Our work has the same flavour of the work by Kovachki et al. (2023) who indeed extends the classical feed forward neural network architecture to function spaces and has required an entire work published in JMLR.
>    Anyhow, we have tried to answer the reviewer's concern by extensively improving the accessibility of multiple parts of the paper (as indicated in blue). In addition, we have also added multiple figures which, we hope, clarify both our architecture and our contribution.
>
> 2. **"Previous methods often lack a general formalism for characterizing their architecture." But there is a well-known paper that does: "Neural Operator: Learning Maps Between Function Spaces. How does the contribution of this paper relate to that one?**
>
>    Indeed, we have cited this influential work multiple times in the submitted version. In this work, the authors have shown that each operator layer maps each hidden representation to the next via the action of the sum of a local linear operator, a non-local integral kernel operator, and a bias function, composing the sum with a pointwise activation function. In particular, the FNO can be covered by a peculiar instance of the integral kernel operator. Our contribution differs by viewing the action of the operator layer as the minimizer of a regularized optimization problem, comparing the current hidden representation and the next. In our case, the choice of the regularization is implicitly linked to the resulting activation function. We also show that we can cover classical neural operators such as FNOs. To summarize, while in the mentioned work the authors have proposed and studied a parametric form of operator layers, our work takes a step back and shows how these parametric forms can notably be seen as minimizers of optimization problems. This perspective allows using the extensive literature on proximal optimization and monotone operator theory to further study the properties of neural operators (see our answers below).
>
> 3. **"In this work, we propose a novel expressive framework for neural operators by conceptualizing the action of operator layers as the minimizers of Bregman regularized optimization problems over Banach function spaces." This sentence does not make sense, please clarify.**
>
>    We are sorry if our sentence is not clear. We wanted to highlight in part the elements provided in the previous answer. Put simply, we would like to emphasize that, in our paper, we demonstrate that applying one operator layer to an input function is equivalent to finding the minimizer of a regularized optimization problem over functions. Our setting requires working in Banach space, and the term expressive refers to the fact that we can cover classic neural operators and multiple activation functions. Please note that, in the revised version (see parts in blue), we have reworked the Contributions paragraph to better clarify the present work. In addition, we have added Figure 3 to help the reader to better understand our framework.

---

> > ### Author Response · Authors · 2024-11-26
> >
> > 4. **"We prove universal approximation results." This is true of almost any reasonable neural network, including MLP. How does this argument show anything special about this particular architecture?**
> >
> >    The fact that any reasonable neural network features universal approximation results is indeed a minimal requirement that anyhow needs to be proved mathematically. Otherwise, contrary results would preclude their use. In addition, without such results, their practical applicability could be questioned.
> >    The proof of universal approximation was done for a number of neural network architectures but was missing for Bregman neural networks (even finite dimensional). In our work, we address the expressivity of such new architecture both in finite and infinite dimension, so to provide more ground to the Bregman neural networks first proposed in Frecon et al. (2022), as well as to Bregman neural operators proposed in our work. The proof itself also allows illustrating the technical aspects, pertaining to our architecture, that must be addressed to provide approximation guarantees. We finally note that, at the moment, our result covers only sigmoidal-type activations. We hope that this work may trigger additional research to possibly extend our approximation results to more general activation functions.
> >
> > 5. **"The proposed framework allows applying the extensive body of literature on proximal numerical optimization, of which Bregman proximity operators belong to, in order to study neural operators." Convince me why this is useful.**
> >
> >    This is useful for multiple reasons, as highlighted in the conclusion, where we have given two examples. First, it permits to provide asymptotic properties, as the number of layers tends to infinity (Combettes & Pesquet, 2020a). Second, one can yield quantitative insights into their stability properties (Combettes & Pesquet, 2020b).
> >    Overall, establishing connections with monotone operator theory, of which Bregman proximity operators belong to, offers powerful tools for studying neural operators with finite width. We hope that our additional explanations have resolved the reviewer's concerns about the research potentials offered by our work. If there are any additional concerns that might affect the scoring of our paper, we would welcome the chance to address them as well.
> >
> > 6. **In Fig 2a, did the same results hold for the other 6 equations solved, or just for this one? Why did you just present the results for Burgers equation?**
> >
> >    The same result holds for all the other equations considered (see Table 2), as well as other architectures, such as WNOs. The corresponding full results (i.e., up to 64 layers) are reported in section D.2 of the appendix. Therefore, we have judged sufficient to only display one result, in the form of a bar plot, in the main paper. If the reviewer finds it helpful, we are open to moving some of the more technical details to the appendix, allowing us to present additional results in the main body of the paper.

---

> > > ### Author Response · Authors · 2024-12-02
> > >
> > > Dear Reviewer,
> > >
> > > The discussion phase ends soon, please let us know if you are convinced by our answer or if you need further information.
> > >
> > > We thank you again for the time dedicated to review our paper and the feedback provided.
> > >
> > > Respectfully,

---

### Official Review · Reviewer_HY9Q · 2024-10-30

**Soundness:** 3
**Presentation:** 2
**Contribution:** 2
**Rating:** 5
**Confidence:** 3

**Summary:**

This paper studies neural operators from the perspective of Bregman proximal optimization. The nonlinear activation layers such as (sigmoid, tanh, SoftPlus) are interpreted as the Bregman proximity operators. Based on the above optimization viewpoint, a new neural operator named BFNO is proposed as an extension of FNO, where an additional term sigma^{-1}(.) introduced in the expression of FNO before the function sigma(). A few experiments show that BFNO performs better than FNO especially for a large number of neural layers.

**Strengths:**

(1) The main contribution of the paper is that the authors interpret the neural operators as Bregman proximal optimization. This opens up the possibility of bringing knowledge or theory of Bregman proximal optimization into the field of neural operators for many possible future work.

(2) As I mentioned earlier,  BFNO is proposed as an extension of FNO by introducing an additional term sigma^{-1}(.) in the expression of FNO before the function sigma(), which I think has a similar effect as the skip connection in F-FNO or ResNet.

**Weaknesses:**

(1) I think the authors should compare the performance of BFNO to that of F-FNO equipped with skip connection. This is because from a high level point of view, the introduction of an additional term sigma^{-1}(.) in the expression of FNO before the function sigma() is very similar to the skip connection in F-FNO. It is very interesting to find out which one performs better.  Personally I think F-FNO might perform better because the skip connection also has a strong motivation from the ODE point of view.  BatchNormalization can also be included in  F-FNO smoothly. If the authors are able to show that BFNO performs better instead with a good explanation, I would be happy to change my score of the paper.

(2) The paper conducted theoretical analysis but were not able to show in theory why BFNO performs better than FNO, which I think is very critical. Instead, they conduct experimental results to argue the superiority of BFNO. This is also partly the reason for me to suggest the comparison between the performance of BFNO and that of F-FNO.

(3) It is not clear to me if BatchNormalization or Layer normalization can also be covered by the framework of Bregman proximal optimization. The reason I have this concern is that the FNO paper used BatchNormalization in their experiment. I would think doing so improve the training stability. If BatchNormalization cannot be covered by  the framework of Bregman proximal optimization, it suggests the limitations of the framework.

(4) Another weakness is that the activation function needs to be monotonic for it to be invertible. This excludes a few functions such as ReLU and Swish.  I understand that SoftPlus is similar to ReLU but still it suggests the framework has some limitations.

**Questions:**

(1) I wonder if the authors use BatchNormalization when implementing FNO in their experiments because the original FNO paper used it.

---

> ### Author Response · Authors · 2024-11-26
>
> 1. **I think the authors should compare the performance of BFNO to that of F-FNO [...].**
> We sincerely thank the reviewer for their kind comment. Indeed, both approaches exhibit similarities. We highlight that BFNO can also be viewed from the ODE point of view through a change of variables. More precisely, and without loss of generality, let us consider a simplified update of the form $v_{t+1} = \sigma( \sigma^{-1}(v_{t}) + W_t v_{t})$. Then, for $z_{t} = \sigma^{-1}(v_{t})$, it follows that $z_{t+1} = z_{t} + W_t \sigma (z_{t})$ which can be seen as a discretization of $\frac{\mathrm{d}z(t)}{\mathrm{d}t} = W(t)\sigma(z(t))$. In contrast, a residual-based architecture of the form $v_{t+1} = v_{t}+\sigma(W_t v_{t})$, such as F-FNO, would lead to the following ODE $\frac{\mathrm{d}v(t)}{\mathrm{d}t} = \sigma(W(t)v(t))$ on $v$ itself. The fact that, in BFNO, the weight matrix is placed outside the activation function can be seen as a different way of mitigating vanishing gradients compared to residual architectures like F-FNO. Concerning the comments on the performance comparison and on the BatchNormalization, we point the reviewer to our general answer.
>
> 2. **The paper conducted theoretical analysis but were not able to show in theory why BFNO performs better [...]**
> We thank the reviewer for highlighting the importance of a theoretical analysis to explain why BFNO outperforms FNO in practice. We agree that such insights would strengthen the paper. However, to the best of our knowledge, providing a formal proof of architectural superiority is a highly challenging task in the current landscape of theoretical deep learning research. One promising avenue for future work could involve analyzing the expressivity and trainability of BFNO using frameworks such as mean-field theory (e.g., [Ref. 1, Ref. 2]). However, this would require an entirely new work.
> That said, we have incorporated the reviewer’s suggestion to compare BFNO with F-FNO experimentally (see our general answer).
>
> 3. **It is not clear to me if BatchNormalization [...].**
> To the best of our knowledge, both BatchNormalization and LayerNormalization cannot be seen through the prism of proximal optimization techniques. Therefore, such architectures cannot be put in the framework of monotone operator theory. Nonetheless, this does not preclude the practitioner from using such layers both for FNO and BFNO. In the experiments described in our general answer, we have tried this and have found a marginal improvement.
>
> 4. **Another weakness is that the activation function needs to be monotonic [...].**
> Concerning the Bregman variant (e.g., BFNO), our framework requires the activation function to be *strictly* monotone (and so invertible). However, we would like to stress that our framework is more general. For reviewer's convenience, we report here the answer we gave to Reviewer JYsk.
> In section 3.1, we presented the general framework (6) for a layer operator. Looking at such general form, one can note that there is an outer operation (which is the $\mathrm{prox}$) and an inner invertible operation (which is $\tilde{\nabla} \Phi_t$). Therefore, by playing with the choices of the functions $\Phi_t$ and $g_t$ one can cover a number of situations of the form $\sigma_2(\sigma_1^{-1}(v)+\mathcal{K}_t(v)+b_t)$ with $\sigma_1$ being *strictly* monotone and $\sigma_2$ only monotone. Classical and Bregman neural operators emerge as special cases, where i) $\sigma_1 = \mathrm{Id}$ and $\sigma_2$ is any monotone function, for the former, and ii) $\sigma_1 = \sigma_2$ is *strictly* monotone, for the latter. We have made this clearer in the revised Section 3.1 (see the parts in blue).
>
> 5. **I wonder if the authors use BatchNormalization when implementing FNO in their experiments because the original FNO paper used it.**
> For our experiments, we relied on the latest available version of the FNO implementation, which does not include BatchNormalization. We note that the original FNO code was removed from the GitHub repository by its author (the 'master' branch was deleted). While we retrieved an earlier version of the code [https://github.com/zeng-hq/fourier_neural_operator/commit/6f1d6fe8ac0b26a6941135dbc5c77b3c1cfcaf19](https://github.com/zeng-hq/fourier_neural_operator/commit/6f1d6fe8ac0b26a6941135dbc5c77b3c1cfcaf19), we observed that BatchNormalization was implemented in the initial commit but was subsequently removed in a later commit (commit number 6f1d6fe) titled "remove unnecessary batchnorm". We hope this clarifies our implementation choices and are happy to provide further details if needed. We have added an experiment in our general answer to support why BatchNormalization is not necessary.
>
>
> [Ref. 1] T. Koshizuka et al. Understanding the expressivity and trainability of Fourier neural operator: a mean-field perspective, NeurIPS 2024.
>
> [Ref. 2] G. Yang and S. Schoenholz. Mean field residual networks: On the edge of chaos. NeurIPS 2017.

---

> > ### Comment · Reviewer_HY9Q · 2024-11-27
> >
> > I can see that the authors have provided additional experiments. The results are interesting. But it does not show that BFNO outperform F-FNO consistently. For 16 layers, BFNO performs slightly better. But it is not clear why it is better.  Therefore, I will keep my score.

---

> ### Author Response · Authors · 2024-11-28
>
> We thank the reviewer for their prompt response and would like to provide additional context to clarify the reported results, helping them make a more informed decision on the scoring of our paper.
>
> ### 1. BFNO permits to build deep models
> - The key takeaway for practitioners is that **BFNO scales effectively with an increasing number of layers**, ensuring that performance either improves or plateaus. This behavior is driven by two factors. First, the architecture supports stability, as $v_{t+1} = \sigma(\sigma^{-1}(v_t) + W_t v_t) = v_t$ when the weights are zero ($W_t=0$), reducing the layer to an identity mapping. Second, empirical evidence (see Figure 7 on page 29 of the new revision or here https://imgur.com/a/K5PvjOj) shows that many weights converge to (or remain near) zero in deeper models. As a result, when additional layers are unnecessary, they effectively default to the identity, maintaining the model's performance without degradation.
> - It could be argued that, in residual-like networks, the relation $v_{t+1} = v_t + \sigma(W_t v_t) = v_t$ holds when $W_t$ is zero. However, we have not observed cases where the learned $W_t \approx 0$ across all configurations of layers tested. We believe that this discrepancy arises from the underlying dynamics explained by the ODE perspective (see the previous discussion), which makes this behavior challenging to achieve during training in residual networks, unlike in BFNOs.
> - F-FNO was initially designed to support the construction of deeper models (up to 24 layers, whereas BFNO scales to at least 64 layers). However, as demonstrated by our experiments, **this claim does not hold** for the tasks we are considering in our paper: predicting of the final output from the initial condition. We believe their ability to reach 24 layers in their task—predicting all maps from $t$ to $t+1$—relies heavily on additional techniques, such as the Markov assumption and teacher forcing, which are specifically tailored to that type of task. To emphasize this, we present an extended table of results for models with up to 32 layers, clearly showing that **BFNO increasingly outperforms F-FNO as model depth grows**.
> - In practice, cross-validating the number of layers is unnecessary for BFNO. One can simply initialize the model with as many layers as their GPU can handle and proceed with training. This approach guarantees achieving the best possible performance without additional tuning.
>
> ### 2. A more fair comparison with F-FNO
> - In the previously reported results, F-FNO operated under a different setup compared to its competitors, primarily due to two factors: 1. the use of a different lifting layer and 2. operator layers with larger widths. These choices were made to align with the original implementation by its authors. To provide a more aligned comparison and isolate the impact of the operator layers, we present results for a modified model, F-FNO, which uses the same lifting layer and operator width as the other models (i.e., the setting originally considered in the FNO paper). In this setting, **BFNO systematically matches or outperforms F-FNO**.
> - Our initial was to make a comparison with the best F-FNO configuration "out-of-the-box" (see F-FNObase). To achieve the reported performance (see F-FNO), we invested significant effort in carefully tuning the optimizer and cross-validating hyperparameters. In contrast, **BFNO is far simpler to train**, requiring none of the extensive adjustments and optimization tricks necessary for F-FNO.
> - We stress that the main takeaway of our work is not merely to outperform F-FNO, but to demonstrate the robustness and scalability of BFNO in terms of number of layers.
>
> | Setting        | Model            | 4 layers      | 8 layers      | 16 layers      | 32 layers      |
> |----------------|------------------|---------------|---------------|----------------|----------------|
> | **FNO paper**  | FNO              | 13.5 ± 0.1    | 13.0 ± 0.2    | 12.6 ± 0.1     | 12.5 ± 0.1     |
> |                | ResFNO           | **13.5 ± 0.1**| 12.8 ± 0.1    | 13.0 ± 0.1     | 13.4 ± 0.1     |
> |                | F-FNO            | 13.6 ± 0.4    | 12.7 ± 0.2    | 13.0 ± 0.5     | 13.4 ± 0.6     |
> |                | BFNO             | 13.7 ± 0.1    | **12.6 ± 0.1**| **12.2 ± 0.1** | **12.0 ± 0.1** |
> | **F-FNO paper**| F-FNO (base)     | 15.4 ± 0.3    | 14.9 ± 0.4    | 15.1 ± 0.2     | 15.5 ± 0.4     |
> |                | F-FNO            | 13.0 ± 0.1| 12.4 ± 0.1| 12.7 ± 0.2     | 13.4 ± 0.2     |
>
> **Table**: Comparison of Test relative ℓ² error across different architectures and number of layers. *The table in our general answer has been changed accordingly. Changes in the text are indicated in italic*
>
>
>
>
> If any further explanation or additional experiments are needed for a positive reassessment of our work, we would be happy to provide them. Once again, we appreciate the reviewer’s time and valuable feedback.

---

> > ### Author Response · Authors · 2024-12-02
> >
> > Dear Reviewer,
> >
> > The discussion phase ends soon, please let us know if you are convinced by our answer or if you need further information.
> >
> > We thank you again for the time dedicated to review our paper and the feedback provided.
> >
> > Respectfully,

---

### Official Review · Reviewer_HeZd · 2024-11-03

**Soundness:** 3
**Presentation:** 1
**Contribution:** 3
**Rating:** 5
**Confidence:** 2

**Summary:**

This paper proposes a novel expressive framework called BFNO to improve FNO by understanding the action of operator layers via the minimization of Bregman regularized optimization problems.

**Strengths:**

The Bregman-based perspective on neural operators is intriguing, and the paper presents a variety of strong theoretical results.

**Weaknesses:**

1. The writing is poor, and I recommend that the authors carefully revise the paper from start to finish, especially regarding newly defined matrices or functions. For example, in Eq. 4, the definitions of $M_t$ and $K_t$ are not clearly stated when they first appear in the paper.
2. The experiments are too simplistic. The authors only compared BFNO with FNO. I suggest including other FNO improvements as baselines.

typo: the operators in Line 370

**Questions:**

Q: Which $\psi$ in Table 1 do you use in the experiments? Do you compare the BFNOs obtained from different $\psi$?

---

> ### Author Response · Authors · 2024-11-26
>
> 1. **The writing is poor, and I recommend that the authors carefully revise the paper from start to finish, especially regarding newly defined matrices or functions. For example, in Eq. 4, the definitions of $M_t$ and $K_t$ are not clearly stated when they first appear in the paper.**
> We thank the reviewer for their feedback and for identifying the omission regarding the definitions of the matrices in Eq. 4. We have carefully revised the paper to check that all concepts are properly introduced without any missing definition. In addition, we have also improved the writing of multiple parts of the paper and moved the more technical parts to the appendix, as indicated in blue. We would be happy to incorporate any other recommendation that the reviewer deems useful.
>
> 2. **The experiments are too simplistic. The authors only compared BFNO with FNO. I suggest including other FNO improvements as baselines.**
> We appreciate the reviewer’s suggestion to include additional FNO improvements as baselines. Our primary objective was to focus on evaluating the added benefits of the $\sigma^{-1}$ term, which is why we restricted our comparison to the original FNO, a widely used and well-established baseline. That said, based on Reviewer HY9Q's comment, we are now conducting experiments to include comparisons with F-FNOs in the revised version. Preliminary results are provided in our general answer. Additionally, we would like to highlight that, in the submitted version, our experiments also included (in the appendix) multiple datasets and a comparison between WNOs and Bregman WNOs. Finally, following (Takamoto et al., 2022) we also provided a detailed frequency analysis showing high error rates on low frequencies and boundary conditions for FNOs.
>
> 3. **Which $\psi$ in Table 1 do you use in the experiments? Do you compare the BFNOs obtained from different $\psi$?**
> The choice of $\psi$ is directly linked to the choice of the activation function. Since we have resorted to the softplus activation function, this is equivalent to choosing the last $\psi$ of Table 1. The choice of the softplus activation function is motivated by the fact that it serves as a smooth approximation of ReLU, commonly used in FNOs.

---

> > ### Author Response · Authors · 2024-12-02
> >
> > Dear Reviewer,
> >
> > The discussion phase ends soon, please let us know if you are convinced by our answer or if you need further information.
> >
> > We thank you again for the time dedicated to review our paper and the feedback provided.
> >
> > Respectfully,

---

### Official Review · Reviewer_JYsk · 2024-11-05

**Soundness:** 3
**Presentation:** 3
**Contribution:** 3
**Rating:** 8
**Confidence:** 3

**Summary:**

This article constructs neural operators from the point
of view of Bregman optimization problems. The proposed idea uses the dual space of Banach functional theory,
and it allows to recover classical neural operators
and define new ones. Numerical results
on the newly constructed operators improve the accuracy of
state-of-the-art results by using deeper networks.

**Strengths:**

Although technical, the article is well-written and easy to follow.

The contribution raises an important question on the choice
of a metric/divergence in the functional space of the solution u.

**Weaknesses:**

It seems that there is still a gap between the universal approximation result
and the numerical results in the article as
the theoretical assumption about the non-linearity sigma (sigmoid type)
does not hold in the numerical models (sigma=softplus is not sigmoid type).
Therefore it would be good to mention this gap in the conclusion.

**Questions:**

- There is some notation inconsistency in the definition of the kernel K_t^ac in eq. 3
and the K_t in Section 3.1. Are you talking about the same type of kernel in these 2 places?
Why do you use k^(t) as the kernel density, rather than the k_t as before (below eq .3)?

- Section 3.1, it is unclear what the sigma_1 and sigma_2 after Remark 6 comes from,
do they depend on g_t?

- Is bar{D} the closure of the set D in the definition of A in Section 4?
Why do you consider the space C with \bar{D} rather than with D?

- some type in remark 9: no in?

---

> ### Author Response · Authors · 2024-11-26
>
> 1. **It seems that there is still a gap between the universal approximation result and the numerical results in the article as the theoretical assumption about the non-linearity sigma (sigmoid type) does not hold in the numerical models ($\sigma=\text{softplus}$ is not sigmoid type). Therefore it would be good to mention this gap in the conclusion.**
> We agree with the reviewer. We have tried to be careful by mentioning a "*preliminary positive result* concerning the universal approximation properties of Bregman neural operators" in the outline paragraph and at the beginning of Section 4. We propose to reinforce this gap both in the contributions paragraph by adding "*Its applicability is grounded by universal approximation results proven for sigmoidal-type activation operators*" and in the conclusion with "*As part of our theoretical advancements, we have also established universal approximation results for Bregman neural architectures with sigmoidal-type activation functions. However, it must be acknowledged that a gap exists between this result and common practices, which predominantly rely on ReLU-like activations, as in our work, opening the door to new theoretical developments*".
>
> 2. **There is some notation inconsistency in the definition of the kernel $K_t^{\rm ac}$ in eq. 3 and the $K_t$ in Section 3.1. Are you talking about the same type of kernel in these 2 places? Why do you use $k^{(t)}$ as the kernel density, rather than the $k_t$ as before (below eq. 3)?**
> The reviewer is right, we have now made the notation uniform and consistent along the paper, as shown in blue.
>
> 3. **Section 3.1, it is unclear what the $\sigma_1$ and $\sigma_2$ after Remark 6 come from, do they depend on $g_t$?**
> The reviewer is correct. To recall, in section 3.1, we presented the general framework (6) for a layer operator. Looking at such general form, one can note that there is an outer operation (which is the $\mathrm{prox}$) and an inner invertible operation (which is $\tilde{\nabla} \Phi_t$). Therefore, by playing with the choices of the functions $\Phi_t$ and $g_t$ one can cover a number of situations of the form $\sigma_2(\sigma_1^{-1}(v)+\mathcal{K}_t(v)+b_t)$ with $\sigma_1$ being *strictly* monotone and $\sigma_2$ only monotone. Classical and Bregman neural operators emerge as special cases, where i) $\sigma_1 = \mathrm{Id}$ and $\sigma_2$ is any monotone function, for the former, and ii) $\sigma_1 = \sigma_2$ is *strictly* monotone, for the latter. We have made this clearer in the revised Section 3.1 (see the parts in blue).
>
> 4. **Is $\bar{D}$ the closure of the set $D$ in the definition of A in Section 4? Why do you consider the space $C$ with $\bar{D}$ rather than with $D$?**
> The reviewer is right, it indeed corresponds to its closure. This has been made explicit in the revised version. The reason to consider the closure is that you need to evaluate the functions on the boundary of the domain when boundary conditions are imposed. This is pretty standard in the literature on PDEs (see, for instance, the book *Introduction to Partial Differential Equations* by G.B. Folland). By the way, this choice is also made in (Kovachki et al. 2023, Section 9.3).
>
> 5. **Some type in Remark 9: "no in"?**
> Since $\Psi$ is a strongly convex Legendre function, there is a unique solution and the equality holds. Note that in the revised version, we decided to remove it from the main body to gain space for additional comments since this was not a critical part for understanding our contribution.

---

> > ### Comment · Reviewer_JYsk · 2024-11-27
> >
> > Thanks for your answers, which clarified my questions. I would keep my acceptance score since I find the presentation quite clear and the idea interesting. But I decreased my confidence score since I am not an expert on this topic.

---

> > > ### Author Response · Authors · 2024-12-02
> > >
> > > Dear Reviewer,
> > >
> > > Thank you for your follow-up message and for confirming that our responses clarified your questions. We are pleased to hear that you find the presentation clear and the idea interesting.
> > >
> > > We understand and respect your decision regarding the confidence score. Please let us know if there are additional clarifications or details we can provide to further support your evaluation.
> > >
> > > Respectfully,

---

### Author Response · Authors · 2024-11-26
**General Answer (Part 1: Revisions)**

Dear Reviewers,


We sincerely thank you for your thorough evaluations and insightful comments, which have been instrumental in refining our work. **We kindly invite the reviewers to read our revised version**, where all the changes are indicated in blue. Below, we address key points raised across the reviews.

On the theoretical side, Reviewer JYsk highlighted a gap between our universal approximation results and their connection to the experimental setup, regarding the activation functions. Reviewer HY9Q also raised valid comments about the monotonicity assumption for the activation operators, suggesting that we further clarify its implications. In response, we have expanded our theoretical discussion to better connect these aspects, explicitly acknowledging the assumptions made and their influence on the scope of the results. We have also provided more detailed introduction and conclusion paragraphs to contextualize the universal approximation results and suggest avenues for future work.

Regarding clarity and presentation, several reviewers emphasized the importance of improving the accessibility of the paper. To address these concerns, we have moved the more technical parts to the appendix, and we have included two additional figures (Figure 1 and Figure 2). In particular, we believe that Figure 2 could motivate neural operator specialists to explore the connections with convex analysis. In addition, many parts have been reworked to provide more intuitions. **We are confident that these changes will enhance the readability of the manuscript and help make the contributions more accessible to a wider audience** despite its mathematical nature.

In terms of numerical experiments, both Reviewer HY9Q and Reviewer HeZd suggested additional comparisons, such as incorporating baselines like F-FNO and exploring configurations with BatchNormalization. We have initiated further experiments to address these suggestions and we have integrated the results into the revised manuscript in the appendix. **Preliminary results are provided in our second general answer.** Finally, we want to stress a point that Reviewer n7Dc may have missed: our novel architecture has already been tested on multiple datasets (Table 2), with detailed results provided in the appendix (Section D). While space constraints prevent the inclusion of all findings, we could move some results in the main body of the paper if deemed useful.

We hope these improvements and the subsequent answers will address your concerns and encourage a positive reassessment of your evaluation.


Respectfully,

The authors.

---

### Author Response · Authors · 2024-11-26
**General Answer (Part 2: Additional Experiments) [1/2]**

Dear Reviewers,

We once again sincerely thank you for your valuable feedback regarding the experimental aspects. In response, we have conducted additional experiments to address your concerns. Due to time constraints, the additional experiments were carried out on the 2D Navier-Stokes dataset ($\nu = 10^{-4}$) with the rigorous experimental setup described in Appendix C.6. In the final version, we will extend this analysis to all the PDEs considered in the paper.

**1. Comparison with F-FNO and Skip Connections.**

Reviewers HeZd and HY9Q expressed concerns about the lack of comparisons with other FNO improvements. We agree that F-FNO is a particularly relevant baseline for comparison, as it i) incorporates skip-like connections that share similarities with our additional $\sigma^{-1}$ term and ii) also seeks to enable the development of deeper FNO architectures.

We have included a comparison with the best-performing F-FNO model (as identified by its authors), trained using our optimization strategy and adapted to our specific learning task. It is important to note that F-FNO was originally designed for predicting mappings between multiple consecutive time steps (e.g., from $t$ to $t+1$) and it offers the option to rely on techniques such as the Markov assumption and teacher forcing. Since our task involves predicting the final state directly from the initial conditions, those techniques are not appropriate, and thus we did not include them in the implementation. *Hereafter, we refer to **F-FNObase** as the original architecture proposed by the authors, trained using their optimization strategy. Conversely, **F-FNO** denotes a model aligned with our experimental setup, featuring a similar architectural design (identical lifting/projection layers and operator widths) and trained using our optimization approach (see Appendix C.6)*. Results reported in table below show that the "out-of-the-box" model F-FNObase does not transfer well to our learning task, as evidenced by the poor performance across all layers, compared to the other models. On the contrary, with our training strategy and adapted hyperparameters, F-FNO shows strong performance, especially with few layers.

Moreover, to isolate the impact of residual connections from the broader structural modifications introduced by F-FNO, we have also implemented and compared a ResNet-inspired variant of FNO, referred to as ResFNO. We did this to better understand the role of the residual connection. As shown in following table, ResFNO consistently achieves lower training error than FNO, which aligns with the behavior typically observed in ResNet-like architectures. However, it falls short in terms of generalization to unseen data, as reflected in the test error.

In conclusion, BFNO demonstrates superior scalability than its competitors (see for 16 layers), evidencing that the additional $\sigma^{-1}$ term, inherent to our Bregman model, better facilitates the training and generalization of deeper architectures.

| Metric | Model | 4 layers | 8 layers | 16 layers | 32 layers |
|--------|-------|----------|----------|-----------|-----------|
| **Test $\ell^2$ (\%)** | FNO | **13.5 ± 0.1** | 13.0 ± 0.2 | 12.6 ± 0.1 | 12.7 ± 0.1 |
| | ResFNO | **13.5 ± 0.1** | 12.8 ± 0.1 | 13.0 ± 0.1|13.4 ± 0.1|
| | F-FNO$_{\rm base}$ | 15.4 ± 0.3 | 14.9 ± 0.4 |15.1 ± 0.2|15.5 ± 0.1 |
| | F-FNO | 13.6 ± 0.4 | 12.7 ± 0.2 | 13.0 ± 0.5 |13.4 ± 0.6 |
| | BFNO | 13.7 ± 0.1 | **12.6 ± 0.1** | **12.2 ± 0.1** |**12.0 ± 0.1** |

| Metric |Model|4 layers|8 layers|16 layers|32 layers|
|--------|-------|----------|----------|-----------|-----------|
| **Train $\ell^2$ (\%)** | FNO | 4.6 ± 0.2 | 3.6 ± 0.1 | 4.0 ± 0.1| 4.7 ± 0.0|
| | ResFNO | **3.7 ± 0.2** | **2.9 ± 0.1** | **2.8 ± 0.1** | **3.6 ± 0.4**|
| | F-FNO$_{\rm base}$ | 10.4 ± 0.3 | 8.7 ± 0.3 | 7.3 ± 0.1 | 8.0 ± 0.0|
| | F-FNO | 8.4 ± 0.4 | 6.7 ± 0.4 | 6.5 ± 0.8 |6.4 ± 1.2|
| | BFNO | 4.4 ± 0.3 | 3.7 ± 0.3 | 3.4 ± 0.2 | 3.7 ± 0.1|

**2. BatchNormalization Investigation.**

Reviewer HY9Q rightfully mentioned that the original FNO code used BatchNormalization (BN). However, as pointed in our specific answer to the reviewer, the authors of FNO have later removed such layer in their code. Since we have used the latest version, we have also not considered BN.

To complement our results, we have conducted new experiments with BN for both FNO and BFNO architectures. Results, reported in the table below, show marginal improvements for 8-layer models (FNO: 13.0% → 12.8%, BFNO: 12.6% → 12.4%) but no consistent benefits for other configurations. This aligns with the conclusion of the recent FNO implementations that removed BN.

| Architecture | 4 layers | 8 layers | 16 layers |
|--------------|----------|----------|-----------|
| FNO | **13.5 ± 0.1** | 13.0 ± 0.1 | 12.6 ± 0.1 |
| FNO + BN | **13.5 ± 0.1** | 12.8 ± 0.2 | 12.6 ± 0.1 |
| BFNO | 13.7 ± 0.1 | 12.6 ± 0.1 | **12.2 ± 0.1** |
| BFNO + BN | **13.5 ± 0.1** | **12.4 ± 0.0** | 12.3 ± 0.1 |

---

> ### Author Response · Authors · 2024-11-26
> **General Answer (Part 2: Additional Experiments) [2/2]**
>
> **3. Activation Function Analysis.**
>
> Reviewer HY9Q is right to stress that a limitation of our framework is the fact that it requires Bregman variants (such as BFNO) to have a strictly monotonic activation function, which excludes a few functions such as ReLU. This justifies why in our experiments we used Softplus as a surrogate of ReLU.
>
> However, as a thought experiment, we also implemented BFNO with ReLU. The table below shows that BFNO with ReLU achieves comparable or better performance than Softplus for the same number of layers. However, the best results are the same (i.e., 12.2% for 16 layers).
>
> | Architecture | 4 layers | 8 layers | 16 layers |
> |--------------|----------|----------|-----------|
> | FNO (ReLU) | 13.5 ± 0.1 | 13.0 ± 0.1 | 12.6 ± 0.1 |
> | BFNO (Softplus) | 13.7 ± 0.1 | 12.6 ± 0.1 | **12.2 ± 0.1** |
> | BFNO (ReLU) | **13.4 ± 0.2** | **12.2 ± 0.2** | **12.2 ± 0.1** |
>
> We believe these extensive results, both from our original submission and the new additional experiments, validate our theoretical framework while demonstrating its practical benefits. We have carefully designed and conducted all experiments to ensure their rigor and reliability, and we have incorporated these findings in the appendix to address the reviewers’ concerns thoroughly.
>
> We hope these efforts will clarify any remaining uncertainties and provide a stronger basis for a reassessment of our work. We are also happy to provide additional experimental details or conduct further specific comparisons if the reviewers find it helpful.
>
> Respectfully,
>
> The authors.

---

### Note · Authors · 2025-01-29

I have read and agree with the venue's withdrawal policy on behalf of myself and my co-authors.